# Compositional Automata Embeddings for Goal-Conditioned Reinforcement Learning

**Beyazit Yalcinkaya**[*]
University of California, Berkeley
beyazit@berkeley.edu

**Niklas Lauffer**[*]
University of California, Berkeley
nlauffer@berkeley.edu

**Marcell Vazquez-Chanlatte**[*]
Nissan Advanced Technology Center
marcell.chanlatte@nissan-usa.com

**Sanjit A. Seshia**
University of California, Berkeley
sseshia@berkeley.edu

## Abstract

Goal-conditioned reinforcement learning is a powerful way to control an AI agent's behavior at runtime. That said, popular goal representations, e.g., target states or natural language, are either limited to Markovian tasks or rely on ambiguous task semantics. We propose representing temporal goals using compositions of deterministic finite automata (cDFAs) and use cDFAs to guide RL agents. cDFAs balance the need for formal temporal semantics with ease of interpretation: if one can understand a flow chart, one can understand a cDFA. On the other hand, cDFAs form a countably infinite concept class with Boolean semantics, and subtle changes to the automaton can result in very different tasks, making them difficult to condition agent behavior on. To address this, we observe that all paths through a DFA correspond to a series of reach-avoid tasks and propose pre-training graph neural network embeddings on "reach-avoid derived" DFAs. Through empirical evaluation, we demonstrate that the proposed pre-training method enables zero-shot generalization to various cDFA task classes and accelerated policy specialization without the myopic suboptimality of hierarchical methods.

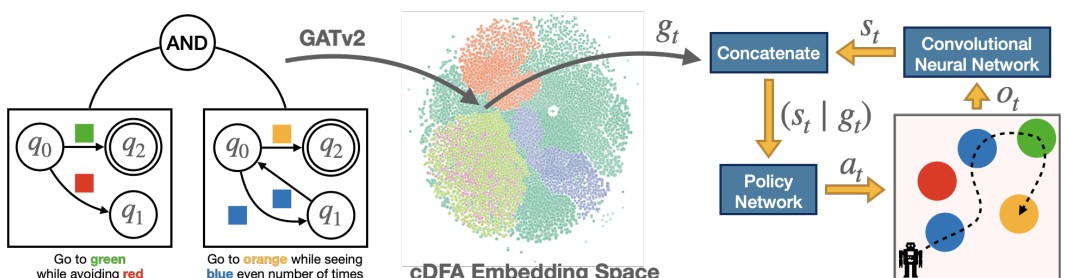

Figure 1: Given a (conjunctive) composition of deterministic finite automata (shown on the left), we construct its embedding using a graph neural network (GATv2) and use this embedding as a *goal* to condition the reinforcement learning policy.

---

[*]Equal contribution

[1]For more information about the project, visit: https://rad-embeddings.github.io/.

38th Conference on Neural Information Processing Systems (NeurIPS 2024).

# 1 Introduction

Goal-conditioned reinforcement learning (RL) [33] has proven to be a powerful way to create AI agents whose task (or goal) can be specified (conditioned on) at runtime. In practice, this is done by learning a goal encoder, i.e., a mapping to dense vectors, and passing the encoded goals as inputs to a policy, e.g., a feedforward neural network. This expressive framework enables the development of flexible agents that can be deployed in a priori unknown ways, e.g., visiting states never targeted during training. The rise of large language models has popularized leveraging natural language as an ergonomic means to specify a task, e.g., "pick up the onions, chop them, and take them to stove."

While incredibly powerful, tasks specified by target states or natural language have a number of shortcomings. First and foremost, target states are necessarily limited to non-temporal tasks. On the other hand, natural language is, by definition, ambiguous, providing little in the way of formal guarantees or analysis of what task is being asked of the AI agent.

To this end, we consider conditioning on tasks in the form of Boolean combinations of deterministic finite automata (DFAs). We refer to this concept class as compositional DFAs (cDFAs). The choice of cDFAs as the concept class is motivated by three observations. First and foremost, DFAs offer simple and intuitive semantics that require only a cursory familiarity with formal languages–*if one can understand a flow chart, one can understand a DFA*. Moreover, recent works have demonstrated that DFA and cDFA can be learned in a few shot manner from expert demonstrations and natural language descriptions [41]. As such, DFAs offer a balance between the accessibility of natural language and rigid formal semantics. The addition of Boolean combinations to cDFA, e.g., perform task 1 AND task 2 AND task 3, provides a simple mechanism to build complicated tasks from smaller ones.

Second, DFAs represent temporal tasks that become Markovian by augmenting the state-space with finite memory. Further, they are the "simplest" family to do so since their finite states are equivalent to having a finite number of sub-tasks, formally Nerode congruences [19]. This is particularly important for goal-conditioned RL which necessarily treats temporal tasks differently than traditional RL. For traditional RL, because the task is fixed, one can simply augment the state space with the corresponding memory to make the task Markovian. In goal-conditioned RL, this is not, in general, possible as *it is unclear what history will be important until the task is provided*. Instead, the encoded task must relay to the policy this temporal information. Third, existing formulations like temporal logics over finite traces and series of reach-avoid tasks are regular languages and thus are expressible as DFAs [11]. This makes DFAs a natural target for conditioning an RL policy on temporal tasks.

The expressivity of DFAs introduces a number of challenges for goal-conditioned RL. First, DFAs form a countably infinite and exponentially growing concept class where subtle changes in the DFA can result in large changes in an agent's behavior. Notably, this means that any distribution over DFAs is necessarily biased with many "similar" DFAs having drastically different probability. Thus, to generally work with DFAs, one cannot simply match finite patterns, but need to learn to encode details necessary for planning. Second, as with traditional goal-based objectives, DFAs provide a very sparse binary reward signal–*did you reach the accepting state or not?* Together with non-trivial dynamics, naïve applications of RL become infeasible due to the lack of dense reward signal. Finally, the exponentially-expanding concept class presents computational limitations for encoding. For example, many interesting DFAs may be too large to be feasibly processed by a graph neural network.

To address these issues of reward sparsity and the need to encode planning, we introduce a distribution of DFAs, called *reach-avoid derived (RAD)*. This concept class is inspired by the observation that all paths through a DFA correspond to a series of (local) reach-avoid problems. We argue in Section 4 that RAD encourages learning to navigate a DFA's structure. Our first key result is that pre-training DFA encoders on RAD DFAs enables *zero-shot generalization* to other DFAs. Next, we treat the problem of DFA size. Many problems are naturally expressed compositionally, e.g., a sequence of rules that must all hold or a set of tasks of which at least one must be accomplished. Due to their Boolean semantics, i.e., did you reach the accepting state or not, DFAs offer well-defined semantics under Boolean compositions. Our second key insight is to encode conjunctions[2] of DFAs (called cDFAs). This is done by using a graph attention network [9] (GATv2) to encode the individual DFA graph structure as well as the conjunction (AND) relationships between a collection of tasks. Recalling that the conjunction of any two DFAs grows (worst-case) quadratically, we observe that cDFAs offer an *exponential* reduction in the size of temporal tasks passed to GATv2.

---

[2]With negations and disjunctions omitted as straightforward extensions.

**1.1 Contributions.** Our main contributions are: **(1)** we propose compositional DFAs (cDFAs), which balance formal semantics, accessibility, and expressivity, as a goal representation for temporal tasks in goal-conditioned RL; **(2)** we propose encoding cDFAs using graph attention networks (GATv2); **(3)** we propose pre-training the cDFA encoder on reach-avoid derived (RAD) DFAs, and **(4)** we perform experiments demonstrating strong zero-shot generalization of pre-training on RAD DFAs.

**1.2 Related Work.** Our work explores a temporal variant of goal-conditioned RL [33] where goals are represented as automata. While traditionally focused on goals as future target states, this has since been extended to tasks such as natural language [7, 21, 26, 37] and temporal logic [38]. While not as expressive as natural language, we believe cDFAs offer a balance between the ergonomics of language while maintaining unambiguous semantics–something of increasing importance due to the seemingly inevitable proliferation of AI agents to safety-critical systems. Moreover, DFA inference has a rich literature [6, 15, 17, 24, 27, 28, 45] with recent works have even shown the ability to learn DFA and cDFA from natural language and expert demonstrations–bridging the gap even more between natural language and automata specified goals [41, 42, 43].

Operating in a similar space, previous work, LTL2Action [38], has shown success with using (finite) linear temporal logic (LTL), a modal extension of propositional logic, to condition RL policies on. In fact, the pre-training and test domains used in this paper are directly derived from that work.

Our choice to focus on DFA rather than LTL is two-fold. First and foremost, over finite traces, LTL is strictly less expressive than DFA. For example, LTL cannot express tasks such as "the number of times the light switch is toggled should be even." Second, like DFA, LTL tasks constitute a countably infinite set of tasks. This again means that any distribution over LTL is necessarily biased to certain subclasses. On the one hand, the syntactic structure makes separation of subclasses very easy. On the other, it remains unclear how to generalize to "common" LTL formula. By contrast, we argue that the local reach-avoid structure of DFAs offers a direct mechanism for generalization. Finally, we note that while LTL is exponentially more succinct than DFA, this is largely mitigated by supporting boolean combinations of DFAs (cDFAs).

Moving away from goal-conditioned RL, recent works have proposed performing symbolic planning on the DFA and cleverly stringing together policies to realize proposed paths [16, 22, 23, 30]. However, their method can still suffer from sub-optimality due to their goal-conditioned value functions being myopic. Specifically, if there are multiple ways to reach the next sub-goal of a temporal task, the optimality of the next action depends on the entire plan, not just the next sub-goal–destroying the compositional structure used for planning. For example, Figure 4 shows a simple example in which hierarchical approaches will find a suboptimal solution due to their myopia. In contrast, conditioning on cDFA embeddings allows our policies to account for the entire task.

On the single-task LTL-constrained policy optimization side, several works have tackled the problem in varying settings [10, 13, 36, 44]. Adjacent to these efforts, various approaches have explored LTL specifications and automaton-like models as objectives in RL [2, 3, 8, 12, 14, 20, 29, 32, 46, 48]. A different line of work considers leveraging quantitative semantics of specifications to shape the reward [1, 22, 25, 36]. However, all of these lines of work are limited to learning a single, fixed goal.

Finally, in a previous work [47], we used hindsight experience replay [5] to solve the reward sparsity problem for DFA-conditioned off-policy learning of DFA task classes. We observe that our RAD pre-training pipeline has similar sample efficiency while generalizing to larger classes of DFAs.

## 2 Preliminaries

In the next sections, we will develop a formalism for conditioning agent behavior on temporal tasks represented as DFAs. To facilitate this exposition, we quickly review goal-conditioned RL and DFA.

**2.1 Goal-Conditioned RL.** We start with technical machinery for goal-conditioned RL.

**Definition 2.1.** A **Markov Decision Process**[3] (MDP) is a tuple $\mathcal{M} = (S, A, T)$, where $S$ is a set of **states**, $A$ is a set of **actions**, and $T$ is a map determining the conditional transition probabilities, i.e. $\Pr(s' \mid s, a) \stackrel{\text{def}}{=} T(s', a, s)$ for $s, s' \in S$ and $a \in A$. We assume that $\mathcal{M}$ contains a sink

---

[3]We omit a reward function since we are in a goal-conditioned setting with varying objectives/rewards.

state, $, denoting the end of an episode. Any MDP, $\mathcal{M}$, can be given $\gamma$-**temporal discounting** semantics by asserting that each transition can end the episode with probability $1 - \gamma$. Formally, $T'(s', a, s) = (1 - \gamma) \cdot T(s', a, s)$, for $s, s' \neq \$$.

**Definition 2.2.** Let $\mathcal{G}$ be a set of states called **goals**. A **goal-conditioned policy**[4], $\pi : S \times \mathcal{G} \to \Delta A$, is a map from state-goal pairs to distributions over actions. Together, an MDP $\mathcal{M}$ and a goal-conditioned policy, $\pi$, define a distribution of sequences of states, called **paths**.

A **goal-conditioned reinforcement learning** problem is a tuple consisting of a (possibly unknown) MDP and a distribution over goals, $\Pr(g)$ for $g \in \mathcal{G}$. The objective of goal-conditioned RL is to maximize the probability of generating a path containing $g$.

### 2.2 Deterministic Finite Automata.
Next, we provide a brief refresher on automata.

**Definition 2.3.** A **Deterministic Finite Automaton** (DFA) is a tuple $\mathcal{A} = (Q, \Sigma, \delta, q_0, F)$, where $Q$ is a finite set of **states**, $\Sigma$ is a finite alphabet, $\delta : Q \times \Sigma \to Q$ is the **transition function**, $q_0 \in Q$ is the **initial state**, and $F \subseteq Q$ are the **accepting states**. The transition function can be lifted to strings via repeated application, $\delta^* : Q \times \Sigma^* \to Q$. One says $\mathcal{A}$ **accepts** $x$ if $\delta^*(q_0, x) \in F$. Finally for simplicity, we abuse notation and denote by DFA the set of all DFA.

Visually, we represent DFAs as multi-graphs whose nodes represent states and whose edges are annotated with a symbol that triggers the corresponding transition. Omitted transitions are assumed to **stutter**, i.e., transition back to the same state. DFAs have a number of interesting properties.

Up to a state isomorphism, all DFAs can be **minimized** to a canonical form, e.g., using Hopcroft's algorithm [18], denoted by minimize($\mathcal{A}$). Further, DFAs are closed under Boolean combinations. For example, the **conjunction** of two DFAs, $\mathcal{A}_1$ and $\mathcal{A}_2$, denoted $\mathcal{A}_1 \wedge \mathcal{A}_2$, is constructed by taking the product of the states, $Q = Q_1 \times Q_2$, and evaluating the transition functions element-wise, $\delta((q_1, q_2), \sigma) = (\delta_1(q_1, \sigma), \delta_2(q_2, \sigma))$. Also, $(q_1, q_2) \in F$ if $q_1 \in F_1$ and $q_2 \in F_2$. While DFAs are closed under conjunction, each pair-wise operation results in a $O(n^2)$ increase in the number of states.

**Definition 2.4.** A **compositional DFA** (cDFA) is a finite set of DFA, $\mathcal{C} = \{\mathcal{A}_1, \ldots, \mathcal{A}_n\}$. The **monolithic** DFA associated to cDFA, $\mathcal{C}$, is given by monolithic($\mathcal{C}$) $\stackrel{\text{def}}{=} \bigwedge_{\mathcal{A} \in \mathcal{C}} \mathcal{A}$. The semantics of a cDFA are inherited from its monolithic DFA, i.e., a string $x$ is accepted by cDFA $\mathcal{C}$ if and only if it is accepted by monolithic($\mathcal{C}$). Graphically, cDFA are represented as trees of depth 1 where the root corresponds to conjunction and each leaf is itself a DFA.

### 2.3 Augmenting MDPs with cDFAs.
Observe that DFAs can be viewed as deterministic MDPs leading to a natural cascading composition we refer to as **DFA-augmention** of a MDP. Informally, one imagines that the MDP and DFA transitions are interleaved. First, the MDP transitions given an action. The new state is then mapped to the DFA's alphabet. This symbol then transitions the DFA. The goal is to reach an accepting state in the DFA before the MDP reaches its end-of-episode state.

Formally, let $\mathcal{A}$ and $\mathcal{M}$ be a DFA and MDP, respectively and let $L : S \to \Sigma$ denote a **labelling** function mapping states to symbols in the DFAs alphabet. The new MDP is constructed as a quotient of the cascading composition of $\mathcal{A}$ and $\mathcal{M}$. First, take the product of the state spaces, $S \times Q$. The action space is $\mathcal{M}$'s action set. The transition of the MDPs state is as before and the DFA transitions are given by: $\delta(q, L(s))$. In order for there to be a unique goal and end-of-episode state, we quotient $S \times Q$ as follows: **(i)** $(\$, q)$ are treated as a single end of episode state and **(ii)** for all accepting states, $q \in F$, the product states $(s, q)$ are treated as a single accepting (goal) state.

## 3 Compositional Automata-Conditioned Reinforcement Learning

We are now ready to formally define (compositional) automata-conditioned RL as a variant of goal-conditioned RL where goals are defined in the augmented MDPs.

**Definition 3.1.** A **DFA-conditioned policy**, $\pi : S \times \text{DFA} \to \Delta A$ is a mapping from state-DFA pairs to action distributions. A **DFA-conditioned RL** problem is a tuple $(\mathcal{M}, P)$, where $\mathcal{M}$ is an MDP and $P$ is a distribution over DFAs. The objective is to maximize the probability of generating a path containing the accepting state. **cDFA-conditioned RL** is defined via the underlying monolithic DFA.

---

[4]Note that in practice, RL (and by extension goal-conditioned RL) policies often consume a state observation rather than the state itself. This is a simple extension to the above which is left out for simplicity of exposition.

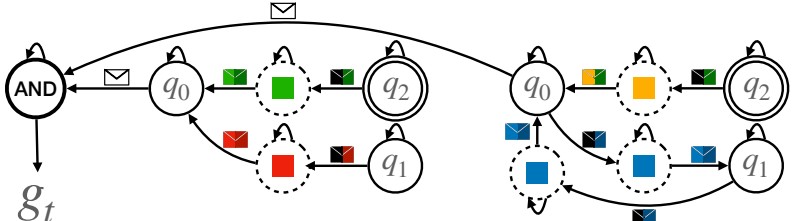

Figure 2: Message passing illustration on the featurization of the cDFA given in Figure 1.

The proposed architecture for conditioning RL policies on temporal tasks encoded as cDFA is given in Figure 1. At step $t$, given an observation $o_t$ and a cDFA $\mathcal{C}_t$, we compute an embedding $s_t$ of the observation $o_t$ using a *convolutional neural network* and embedding $g_t$ of $\mathcal{C}_t$ (constituting a *goal*) using a *message-passing neural network* (MPNN). We then concatenate $s_t$ and $g_t$ and feed it to a policy network to get an action $a_t$. After taking $a_t$, we use the next observation $o_{t+1}$ to update the initial state of each $\mathcal{A}_t \in \mathcal{C}_t$ based on the transition taken. We then minimize each updated DFA to form the next cDFA $\mathcal{C}_{t+1}$, which is used to get the next action $a_{t+1}$.

**3.1 cDFA Featurization.** In order to pass a cDFA $\mathcal{C}_t$ to an MPNN, we construct a featurization $\hat{\mathcal{G}}_{\mathcal{C}_t} = (\hat{\mathcal{V}}_{\mathcal{C}_t}, \hat{\mathcal{E}}_{\mathcal{C}_t}, h_{\mathcal{C}_t})$. Details of this process are given in Appendix A. Essentially, we apply four sequential modifications to the graph structure trivially induced by $\mathcal{C}_t$: (i) add new nodes for every transition in each DFA in the composition, remove these transitions to connect the new nodes to the source and target nodes of the removed transitions, and in the feature vectors of the new nodes, encode all symbols triggering that transition using positional one-hot encodings, (ii) reverse all the edges, (iii) add self-loops to all nodes, and (iv) add edges from the nodes corresponding to the initial states of DFAs to the "AND" node. Figure 2 shows the featurization of the cDFA given in Figure 1.

**3.2 cDFA Embedding.** Given a cDFA featurization $\hat{\mathcal{G}}_{\mathcal{C}_t} = (\hat{\mathcal{V}}_{\mathcal{C}_t}, \hat{\mathcal{E}}_{\mathcal{C}_t}, h_{\mathcal{C}_t})$, we construct an embedding of the cDFA using a *graph attention network* (GATv2) [9] performing a sequence of *message passing* steps to map each node to a vector. For ease of notation, we refer to the features of a node $v \in \hat{\mathcal{V}}_{\mathcal{C}_t}$ as $h_v \stackrel{\text{def}}{=} h_{\mathcal{C}_t}(v)$. At each message passing step, node features are updated as:

$$h'_v = \sum_{u \in \mathcal{N}(v)} \alpha_{vu} W_{tgt} h_u,$$

where $\mathcal{N}(v)$ denotes neighbors of $v$, $W_{tgt}$ is a linear map, and $\alpha_{vu}$ is the attention score between $v$ and $u$ computed as:

$$\alpha_{vu} = \text{softmax}_{\text{i}}(e_{vu}) \qquad e_{vu} = a \cdot \text{LeakyReLU}\left(W_{src} h_v + W_{tgt} h_u\right),$$

where $a$ is a vector and $W_{src}$ is a linear map. After message passing steps[5], the feature vector of the cDFA's "AND" node (which has received messages from all $\mathcal{A}_t \in \mathcal{C}_t$) represents the latent space encoding $g_t$ of the temporal task given by $\mathcal{C}_t$.

## 4 Pre-training on Reach-Avoid Derived (RAD) Compositional Automata

In this section, we introduce pre-training on reach-avoid derived (RAD) cDFAs. This pre-training is designed to solve three important problems. First, we wish to avoid simultaneously learning a control policy and embeddings for the cDFAs. Second, we wish to learn a domain-independent cDFA encoder that performs well across cDFA distributions. Third, we wish to have either a domain-specific or a general cDFA-conditioned policy that performs well across different cDFA distributions.

To alleviate the first problem, i.e., decoupling of learning of control and cDFA embeddings, we follow [38] and pre-train our cDFA encoding GNN on a "**dummy**" MDP containing a single non-end-of-episode state. To solve this dummy MDP, we connect a single linear layer to the output of the GNN which maps cDFA embeddings to the symbols in its alphabet. We then take these symbols and

---

[5]Note that the same weights, $W_{src}$ and $W_{tgt}$, are used in each message-passing step.

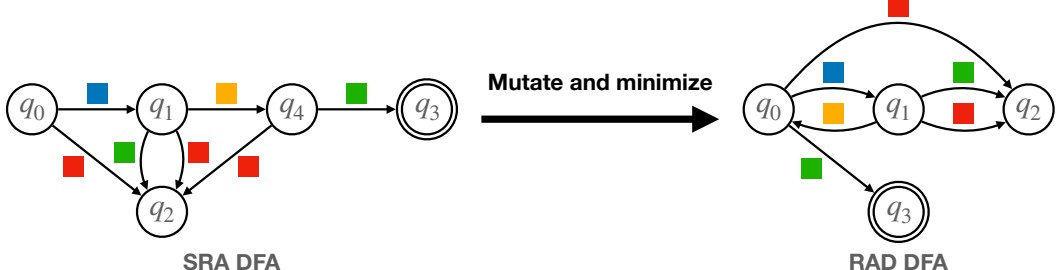

Figure 3: An example of a sequence of local reach-avoid problems and a RAD DFA obtained from it.

take a transition (either a stuttering one or one that changes the state of the DFA) in each DFA of the composition. The episode ends either when all DFAs end up in their corresponding accepting states (reward 1) or when one of the DFAs in the composition ends up in a non-accepting sink state (reward $-1$). The cDFA-augmentation then only models the dynamics of the cDFA being conditioned on. The resulting cDFA-conditioned RL problem can thus focus on only learning to encode the cDFAs.

**4.1    Sequential reach-avoid as the basis for planning.**  What remains then is to support generalization to other cDFA distributions. In particular, recall that cDFAs (and DFAs) lack a canonical "unbiased" distribution making it a priori difficult to pre-select a cDFA distribution. To begin, observe that realizing any individual state transition $q \to q'$ through a DFA corresponds to a **reach-avoid** task: **(i)** eventually transition to state $q'$ and **(ii)** avoid symbols that transition to a state different than $q$ and $q'$. Further, a path through a DFA, $q_1, \ldots, q_n$, corresponds to a sequence of reach-avoid (SRA) tasks. Notably, as illustrated in Figure 3, SRA tasks can be represented as DFA. Thus, in order to generally reason about satisfying a DFA, our neural encoding must be able to solve SRA tasks.

**4.2    Reach-avoid derived DFAs.**  For general DFAs, there can be arbitrarily many (often infinitely many) paths leading to an accepting state, resulting in multiple SRA tasks interacting and interleaving. Thus, the overall structure of a DFA might be much richer than an SRA task even though it consists of local reach-avoid problems. Building on the observation that paths of a DFA correspond to SRA tasks, we define reach-avoid derived DFAs as a generalization of SRA.

**Definition 4.1.** We define a **mutation** of a DFA $\mathcal{A}$ to be the process of randomly changing a transition of $\mathcal{A}$, removing the outgoing edges of the accepting state, and then minimizing. To change a transition, we uniformly sample a state-symbol-state triplet $(q, \sigma, q')$ and define $\delta(q, \sigma) \stackrel{\text{def}}{=} q'$. A DFA $\mathcal{A}$ is said to be $m-$**reach-avoid derived** ($m$-RAD) if it can be constructed by mutating an SRA DFA $m$-times.

We define the $(m, p)-$**RAD distribution** over DFAs as follows: **(i)** sample a SRA DFA, $\mathcal{A}$, with $k+2$ states where $k \sim \text{Geometric}(p)$; each transition stutters with probability 0.9 and is otherwise made a reach or avoid transition with equal probability; **(ii)** sequentially apply $m$ mutations where mutations leading to trivial one-state DFAs are rejected. To avoid notational clutter, we shall often suppress $m, p$ and say RAD DFAs. Finally, we define the RAD cDFA distribution by sampling $n$ RAD DFAs, where $n$ is also drawn from a geometric distribution. See Appendix B for the pseudocode. Figure 3 demonstrates how mutations and minimization can turn an SRA DFA into a RAD DFA. To get the RAD DFA, we apply two mutations to the SRA DFA, adding two new transitions: (i) from $q_0$ to $q_3$ on green and (ii) from $q_4$ to $q_1$ on blue, and then the minimization collapses $q_0$ and $q_4$ to a single state.

# 5    Experiments

We explore the following 6 questions on several task classes in discrete and continuous environments:

**RQ1**    Do our policies overcome the limitations of hierarchical approaches?
**RQ2**    Does pre-training of the cDFA encoder on RAD cDFAs accelerate cDFA-conditioned RL?
**RQ3**    Does freezing the cDFA encoder negatively impact cDFA-conditioned RL performance?
**RQ4**    Do the cDFA-embeddings pre-trained on RAD generalize to other task classes?
**RQ5**    How does the RAD pre-trained cDFA encoder represent the corresponding cDFA?
**RQ6**    Do RAD pre-trained cDFA-conditioned **policies** generalize across task classes?

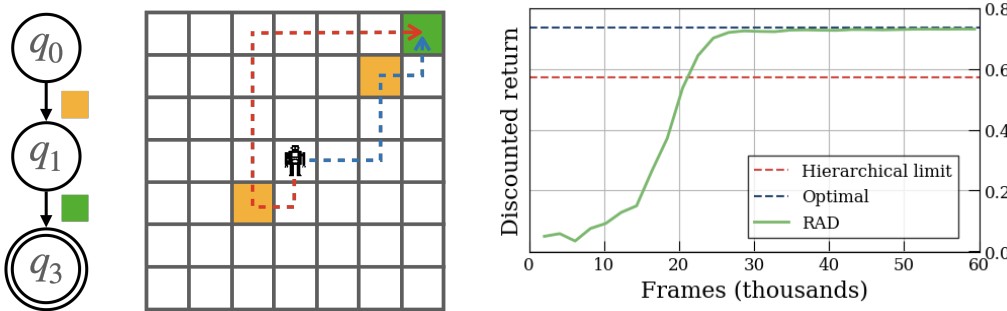

Figure 4: An example in which the myopia of hierarchical approaches causes them to find a suboptimal solution. If the task is to first go to orange and then green, the hierarchical approaches will choose the closest orange which takes them further from green whereas our approach finds the optimal solution.

**Letterworld Environment.** Introduced in [4, 38], Letterworld is a 7x7 grid where the agent occupies one square at a time. Some squares have randomly assigned letters which cDFA tasks are defined over. The agent can move in the four cardinal directions to adjacent squares. Moving into a square with a letter satisfies the corresponding symbol. The agent observes its position and the letters in each grid. Each layout has 12 unique letters (each appearing twice) and a horizon of 75.

**Zones environment.** Introduced in [38] (building on Safety Gym [31]), Zones is as an extension to Letterworld with continuous state and action spaces. There are 8 colored (2 of each color) circles randomly positioned in the environment. cDFA tasks are defined over colors. The agent's action space allows it to steer left and right and accelerate forward and backward while observing lidar, accelerometer, and velocimeter data. The environment has a horizon of 1000.

**Task classes.** In addition to RAD, we consider the following task classes in our experiments.

- **Reach (R):** Defines a total order over $k \sim \text{Uniform}(1, K)$ symbols, e.g., $q_0$ and $q_1$ in Figure 1.
- **Reach-Avoid (RA):** Defines an SRA task of length $k \sim \text{Uniform}(1, K)$, e.g., first DFA in Figure 1.
- **Reach-Avoid with Redemption (RAR):** Similar to RA, but they have transitions from their avoid state to the previous state, e.g., the sub-DFA with states $q_0, q_1, q_3$ in Figure 3.
- **Parity:** A sub-class of RAR where the symbol that leads to the avoid state also triggers a transition back to the previous state, e.g., the DFA on the right in Figure 1. These tasks are not expressible in LTL due to their languages not being star-free regular [40].

Given a bound $N$, using these task classes, we generate their compositional counterparts by sampling $n \sim \text{Uniform}(1, N)$ many DFAs to form a cDFA. As a short-hand notation, for example, we refer to RAR cDFAs with a maximum number of conjunctions 3 where each DFA has a task length of also 2 as cRAR.1.2.1.3 and refer to its associated monolithic (see Definition 2.4) class as RAR.1.2.1.3, i.e., without the "c" prefix. We use the same abbreviation convention for other task classes as well. We use cRAD for RAD cDFAs with truncated distributions and NT-cRAD for the not-truncated version.

Finally, for comparisons with [38], we include the task classes defined in that work, i.e., **partially ordered** (PO) and **avoidance** (AV) finite LTL tasks, represented as cDFAs.

**Pre-training procedure.** We pre-train GATv2 on RAD cDFAs in the dummy MDP as described in Section 4. We set both geometric distribution parameters of the RAD cDFA sampler to $0.5$ and truncate these distributions so that $n \leq 5$ and $2 + k \leq 10$, i.e., we make sure that the maximum number of DFAs in a composition is $5$, and the maximum number of states of each DFA is $10$. As a baseline for GATv2, we also pre-train a *relational graph convolutional network* (RGCN) [34].

**Training procedure.** We experiment with training policies three ways: Training without pre-training (*no pretraining*), with frozen MPNNs pre-trained on RAD as described in Section 4 (*pretraining (frozen)*), and allowing the pre-trained MPNN to continue receiving gradient updates (*pretraining*).

We use proximal policy optimization (PPO) [35] for all reinforcement learning setups, giving a reward of $+1$ when a rollout reaches the accepting state of the cDFA task, $-1$ when a rollout reaches

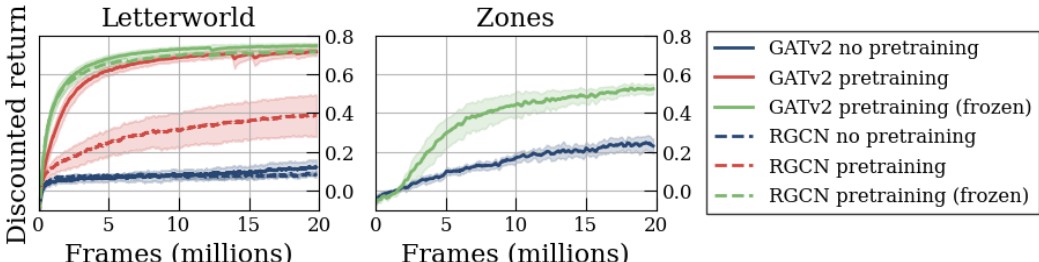

Figure 5: Training curves (error bars show a 90% confidence interval over 10 seeds) in Letterworld (discrete) and Zones (continuous) for policies trained on RAD cDFAs, showing that frozen pre-trained cDFA encoders perform better than non-frozen ones while no pre-training barely learns the tasks.

a rejecting sink state of the cDFA task, and 0 if the environment reaches the timeout horizon without achieving the cDFA task. See Appendix D for the hyperparameters and compute used.

**RQ1 cDFA-conditioned policies do not suffer from myopia.** First, we present the experiment results highlighting the myopia of hierarchical approaches. To this end, we construct a simplified variation of the Letterworld environment, given in Figure 4, where the task is to visit the orange square before the green one. Observe that any hierarchical approach, like [23, 30], based on high-level planning over some automaton-like structure suffers from myopia. Specifically, such approaches do not account for the overall task while planning for the intermediate goals. In this example, hierarchical approaches fail to differentiate between the two orange squares in the context of the given task. On the other hand, our approach quickly converges to the optimal policy as given in Figure 4.

**RQ2,RQ3: Pretraining helps policies perform better faster.** Figure 5 shows training curves for learning on RAD tasks using RGCN (dashed curves) and GATv2 (solid curves) as the MPNN module in Letterworld. Without pre-training in the dummy environment (blue curves), the policies barely learn how to accomplish the tasks, likely because of the difficulty in trying to simultaneously learn an embedding of the cDFA task and learn the dynamics of the environment required to reach certain symbols. Frozen pre-training (green curve) helps tremendously in learning the RAD task class in the Letterworld regardless of the MPNN module architecture. The policies learned are able to satisfy sampled RAD tasks on average over 95% of the time and within 5 timesteps. Non-frozen pre-training (red curve) also helps the policies learn more quickly, but with a significant difference between GATv2 and RGCN. Notably, non-frozen pre-training performs significantly worse for RGCN than its frozen counterpart, showing that freezing the MPNN module after pre-training actually helps performance. We further note that the wall clock training time was significantly faster due to not propagating gradients through the MPNN (about 20% faster).

**RQ4: Pretrained cDFA embeddings generalize well.** We tested pre-trained GATv2 and RGCN models on various new cDFAs and their associated monolithic task distributions in the dummy DFA environment. Figure 10 in Appendix C.2 presents experiment results averaged over 10 seeds and 50 episodes per seed. The left y-axis illustrates the satisfaction likelihood while the right y-axis presents the number of steps needed to complete the task. The figure shows that GATv2 achieves approximately 90% accuracy across all tasks, with nearly 100% accuracy in most cases. Although RGCN generalizes comparably to GATv2 in most instances, it achieves around 80% accuracy for both cRA.1.5.1.5 and cRA.1.1.1.10 whereas GATv2 performs at nearly 90% and 100%, respectively.

Considering that the maximum number of DFAs in a composition seen during training is 5 (sampled with low probability) and the DFA with the maximum task length is 9 (with a maximum of 10 states, sampled with even lower probability), these generalization results are quite remarkable. GATv2 can generalize to 10 DFAs in composition as well as DFAs with a task length of 10 with nearly 100% accuracy. Furthermore, Figure 10 shows that the embeddings produced by GATv2 lead to shorter episodes compared to RGCN, indicating that GATv2 learns better representations for cDFAs. Overall, regardless of the MPNN model used, pre-training on the RAD cDFAs provides generalization.

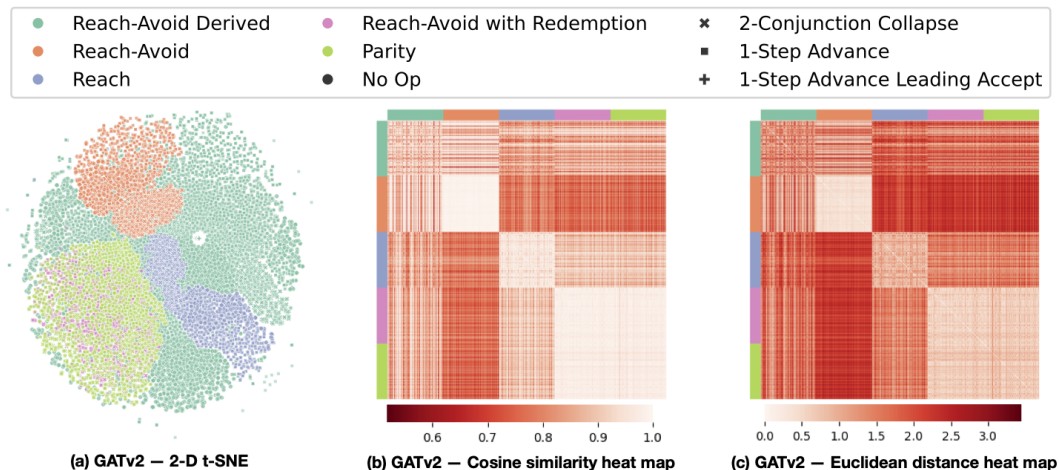

(a) GATv2 — 2-D t-SNE     (b) GATv2 — Cosine similarity heat map     (c) GATv2 — Euclidean distance heat map

Figure 6: Visualizations of the embeddings generated by GATv2 pre-trained on RAD cDFAs, illustrating that the learned embedding space reflects the similarities between different task classes.

**RQ5: cDFA embedding space is well-clustered.** Our GATv2 cDFA encoders learn 32-dimensional embeddings, and, as shown in the previous answer, generalize well to other cDFA classes. To study the cDFA embeddings, we sample cDFAs from RAD, RA, RAR, reach, and parity task classes. For each class, we also introduce two variants: *2-conjunction collapse* and *1-step advance*. The former randomly selects two DFAs of a cDFA and collapses them down to a single DFA, and the latter takes a random non-stuttering transition on the cDFA. We study the embedding space in two different ways. One projects the embeddings down to 2D using t-SNE [39]. The other computes pairwise cosine similarities and Euclidean distances of the embeddings. See Appendix C.3 for the RGCN results.

The results are given in Figure 6. We first observe that modifications like 2-conjunction collapse and 1-step advance result in similar embeddings, showcasing the robustness provided by pre-training GATv2 on RAD. Moreover, due to the 1-step advance operation, some cDFAs end up in their accepting states, and we see that accepting cDFAs are well-isolated inside the green region in Figure 6a. Similarly, we observe that parity samples are mapped very close to RAR samples (its parent class).

The same figure also demonstrates that GATv2 successfully separates different cDFA classes into distinct clusters. This separation is further confirmed by the cosine similarity (c.f. Figure 6b) and Euclidean distance heat maps (c.f. Figure 6c). On the other hand, both heat maps also show that the RAD cDFA class is very rich in terms of various DFAs it contains since it has both similar and different samples compared to other task classes. Notably, we see that the vast majority of RAD cDFA we sample are out-of-distribution for the other classes implying robust generalization.

**RQ6: Policies Trained on RAD generalize well.** Figure 7 (blue) shows how well the policies trained on RAD generalize to other task distributions. Both frozen and non-frozen (see Appendix C.4) pre-trained policies generalize well across the board with a slight edge to frozen pre-training. Notably, the policies suffers almost no performance loss on non-truncated RAD tasks even though it was only trained on the truncated distribution. The policy sees the largest dip in satisfaction likelihood on

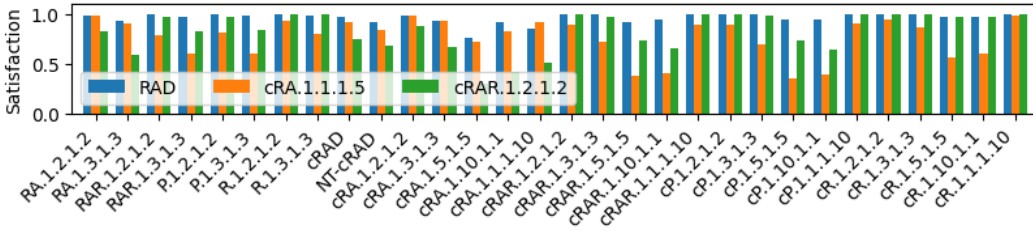

Figure 7: Pre-training the cDFA encoder lets policies trained in Letterworld on subclasses to generalize OOD. All policies were trained using frozen GATv2. See Appendix C.5 for training curves.

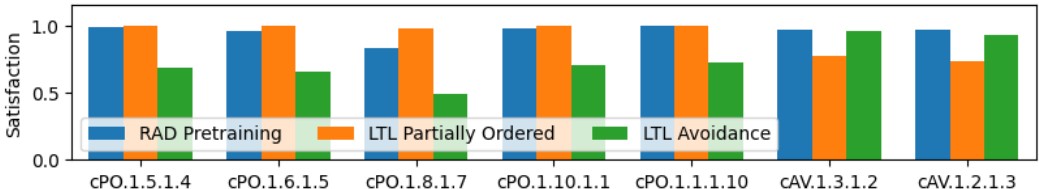

Figure 8: Satisfaction generalization capabilities on LTL tasks (from [38]) of LTL2Action [38] policies vs policies trained on RAD cDFAs. See Appendix C.8 for training curves of policies.

longer compositional reach avoid (cRA) tasks. Interestingly, even very long cRA tasks of up to 10 symbols can be satisfied above 90% of the time, even though such long tasks put the satisfying state in cDFA past the depth that the message passing in our MPNNs reach. Overall, the performance of the non-pre-trained models (see Appendix C.4) follows a similar (albeit dramatically worse) trend as the pre-trained models. They do see some success in the easier task classes such as reach and parity but perform poorly in more difficult task classes that contain rejecting sink states. Our generalization experiments comparing GATv2 to RGCN show that even though RGCN performs comparably, GATv2 outperforms it in all cases (see Appendix C.6).

Figure 7 shows that RAD pre-training allows policies trained in Letterworld to zero-shot generalize far out of distribution. If you take a RAD pre-trained cDFA encoder and train a policy on some small subclass (cRA.1.1.1.5 and cRAR.1.2.1.2), the policies can still handle much longer tasks within their class and even tasks outside of their class. In the case of cRA.1.1.1.5, the training of the policy module never gets to see a multistage task, only multiple conjunctions, yet it's able to satisfy tasks from the full RAD distribution 90% of the time.

We also compare against policies trained using LTL syntax embeddings [38]. Figure 21 in Appendix C.8 shows training curves for these policies where pre-training was performed similar to RAD pre-training except only on the specialized class that randomly samples between PO.1.5.1.4 and AV.1.3.1.2 (task classes from [38]). Figure 8 shows how well these policies and our RAD pre-trained policy generalize in the Letterworld to harder version of PO and AV. The results show that our method generalizes well across all tasks. Furthermore, in some cases, it outperforms the LTL policy on the task it is specifically trained on, e.g., even though the LTL policy shown by the green bar is trained on cAV.1.3.1.2, our method outperforms it. Similarly, observe that although our method was not specifically trained on cPO.1.5.1.4 tasks, it achieves 100% accuracy just like the LTL policy that is specifically trained on this task class. In addition to all of the earlier generalization results, this figure shows that RAD pre-training generalizes to these out of distribution tasks just as well as the policies from [38] even though it is not specifically trained on these tasks.

## 6 Conclusion

We introduced compositional deterministic finite automata (cDFA) as an expressive and robust goal representation for temporal tasks in goal-conditioned reinforcement learning (RL). Addressing the limitations of hierarchical approaches and the challenge of extending goal-conditioned RL to non-Markovian tasks, we proposed using cDFAs to leverage formal semantics, accessibility, and expressivity. By employing graph attention networks (GATv2) to construct embeddings for cDFA tasks and conditioning the RL policy on these representations, our method encourages the policy to take the entire temporal task into account during decision-making. Additionally, we introduced reach-avoid derived (RAD) cDFAs to enable the learning of rich task embeddings and policies that generalize to unseen task classes. Our experiments show that pre-training on RAD cDFAs provides strong zero-shot generalization and robustness. Using cDFAs to encode temporal tasks enhances explainability and enables automated reasoning for interpretable and safe AI systems.

**6.1 Limitations.** A limitation of this work is the assumption of a labeling function mapping MDP states to the task alphabet. Also, further research is needed to understand *how* cDFA-encoders learn to represent tasks and how much further their generalization capabilities can be pushed.

## Acknowledgments and Disclosure of Funding

This work is partially supported by DARPA contracts FA8750-18-C-0101 (AA) and FA8750-23-C-0080 (ANSR), by Nissan and Toyota under the iCyPhy Center, and by C3DTI. Niklas Lauffer is supported by an NSF graduate research fellowship.

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

# A  Details of cDFA Featurization

To construct the featurization of a cDFA $\mathcal{C}_t$, we first build individual graph featurizations for each DFA in the composition and then combine them in a single featurization for $\mathcal{C}_t$. For each $\mathcal{A}_t \in \mathcal{C}_t$, we construct a directed graph $\mathcal{G}_{\mathcal{A}_t} = (\mathcal{V}_{\mathcal{A}_t}, \mathcal{E}_{\mathcal{A}_t}, \mathcal{P}_{\mathcal{A}_t})$, where $v_i \in \mathcal{V}_{\mathcal{A}_t}$ represents the states of $\mathcal{A}_t$, $e_{vu} = (v, u) \in \mathcal{E}_{\mathcal{A}_t}$ represents the transitions of $\mathcal{A}_t$, and $\mathcal{P}_{\mathcal{A}_t} : \mathcal{E}_{\mathcal{A}_t} \mapsto 2^{\Sigma_t}$ maps each edge $e_{vu}$ to a set of symbols in $\mathcal{A}_t$'s alphabet, denoting different symbols triggering a transition from $v$ to $u$. Using the graph representation $\mathcal{G}_{\mathcal{A}_t}$, we construct the DFA featurization $\hat{\mathcal{G}}_{\mathcal{A}_t} = (\hat{\mathcal{V}}_{\mathcal{A}_t}, \hat{\mathcal{E}}_{\mathcal{A}_t}, h_{\mathcal{A}_t})$. The sketch of the iterative construction of $\hat{\mathcal{G}}_{\mathcal{A}_t}$ is given in the following paragraph.

Initially, $\hat{\mathcal{G}}_{\mathcal{A}_t}$ has all the nodes of $\mathcal{G}_{\mathcal{A}_t}$, and it does not have any edges. For each $e_{vu} = (v, u) \in \mathcal{E}_{\mathcal{A}_t}$, add a new node $e_{vu}$ to $\hat{\mathcal{V}}_{\mathcal{A}_t}$ and two new edges to $\hat{\mathcal{E}}_{\mathcal{A}_t}$ – one from $v$ to the new node $e_{vu}$ and another one from $e_{vu}$ to $u$. Then, add self-loop edges for each node. Once $\hat{\mathcal{G}}_{\mathcal{A}_t}$ is constructed this way, we reverse its edges to ensure that message passing transmits information to the initial state of the DFA. To include the details of individual nodes, each $v \in \hat{\mathcal{V}}_{\mathcal{A}_t}$ is associated with an input node feature $h_{\mathcal{A}_t}(v)$. For $v \in \mathcal{V}_{\mathcal{A}_t}$, $h_{\mathcal{A}_t}(v)$ indicates whether $v$ is the initial, an accepting, or a rejecting state. For each $e_{vu} \in \hat{\mathcal{V}}_{\mathcal{A}_t} \setminus \mathcal{V}_{\mathcal{A}_t}$, $h_{\mathcal{A}_t}(e_{vu})$ has one-hot positional encodings of each symbol in $\mathcal{P}_{\mathcal{A}_t}(e_{vu})$.

Once all featurizations are constructed for each $\mathcal{A}_t \in \mathcal{C}_t$, we construct the composition's featurization $\hat{\mathcal{G}}_{\mathcal{C}_t} = (\hat{\mathcal{V}}_{\mathcal{C}_t}, \hat{\mathcal{E}}_{\mathcal{C}_t}, h_{\mathcal{C}_t})$. It is constructed by taking the union of all $\hat{\mathcal{G}}_{\mathcal{A}_t}$ for each $\mathcal{A}_t \in \mathcal{C}_t$, adding an "AND" node $v_{\text{AND}}$ to the graph, and including new edges from the nodes corresponding to the initial states of all $\mathcal{A}_t \in \mathcal{C}_t$ to $v_{\text{AND}}$. Notice that taking this union is trivial since all nodes and edges of a DFA in the composition are unique. Through one-hot encoding of a unique index in the feature vector $h_{\mathcal{C}_t}(v_{\text{AND}})$ of $v_{\text{AND}}$, the "AND" node is encoded as a different kind of node that does not share any features with other nodes in the graph, making it distinguished compared to nodes representing DFA states and transitions. Figure 2 shows the featurization of the cDFA in Figure 1.

# B  RAD cDFA Sampler Algorithm

Algorithm 1 shows the pseudocode for RAD cDFA sampling.

---

**Algorithm 1** RAD cDFA Sampler

---

**Require:** Geometric distribution parameters $p_n$ and $p_k$ and maximum number of mutations $m$
1:  $n \sim Geometric(p_n)$
2:  $\mathcal{C} = \{\}$
3:  **for** $i \leftarrow 1$ to $n$ **do**
4:      $k \sim Geometric(p_k)$
5:      $\mathcal{A} \leftarrow$ Sample a reach-avoid DFA with $k + 2$ states by sampling a reach and an avoid symbol uniformly from the alphabet for each state and adding these random transitions; for other symbols, add a stuttering transition with probability 0.9, or, make it reach or avoid with equal probability
6:      **for** $i \leftarrow 1$ to $m$ **do**
7:          $\mathcal{A}' \leftarrow \mathcal{A}$
8:          Mutate $\mathcal{A}'$ by sampling a state pair $(s, s')$ and a symbol $a$ and adding $s \xrightarrow{a} s'$
9:          Make $\mathcal{A}'$'s accepting state a sink by removing its out transitions and minimize $\mathcal{A}'$
10:         **if** $\mathcal{A}'$ is **not** a trivially accepting **then** $\mathcal{A} \leftarrow \mathcal{A}'$
11:     $\mathcal{C} \leftarrow \mathcal{C} \cup \{\mathcal{A}\}$
12: **Return** $\mathcal{C}$

---

In the experiments, we bounded the size of sampled cDFAs using truncated geometric distributions for $n$ and $k$, with upper bounds of 5 and 10, respectively, and $k$ was fixed per rollout environment. Using multiple environments with different sampler instantiations, combined with mutations applied to the sampled DFAs, allowed policies to encounter a diverse range of DFAs of varying sizes during training.

# C   Results

**C.1    Pretraining learning curves: GATv2 vs RGCN.**   Figure 9 presents the learning curves for
pre-training of GATv2 and RGCN on RAD cDFAs in the dummy MDP. The figure illustrates that
GATv2 converges faster than RGCN, potentially highlighting the benefits of the attention mechanism.

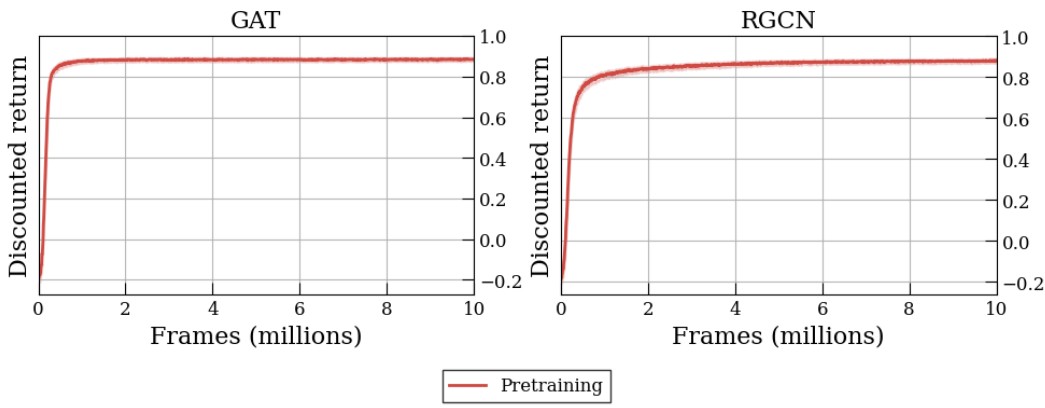

Figure 9: Learning curves for pre-training on RAD cDFAs in the dummy MDP.

**C.2 Generalization of pre-trained cDFA embeddings: GATv2 vs RGCN.** Figure 10 presents the generalization experiment results in the dummy MDP. It shows that pre-trained GATv2 consistently outperforms RGCN with almost 100% accuracy in most cases. It also learns to solve the dummy MDP with less steps compared to RGCN. Overall, we observe that pre-training on RAD cDFAs provides good generalization to other cDFA task classes for both GATv2 and RGCN.

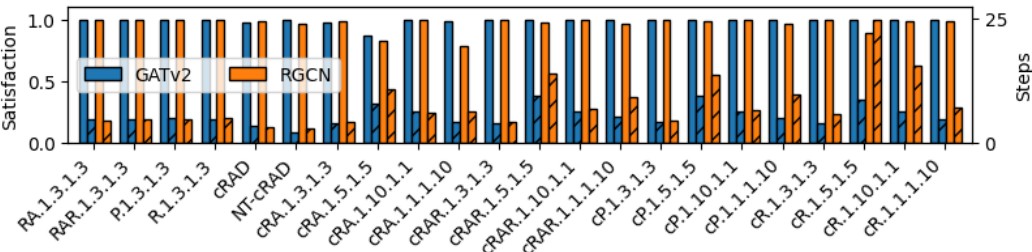

Figure 10: Generalization results of GATv2 and RGCN pre-trained on RAD cDFAs, where satisfaction likelihood (left) and step count (right) are shown by the solid lines and by cross-hatching, respectively.

**C.3  Embedding space analysis: GATv2 vs RGCN.**  Figure 11 presents the embedding space analysis results for both GATv2 and RGCN. Results show that RGCN demonstrates competence in effectively separating different classes, albeit with certain limitations. While it succeeds in distinguishing reach from other classes, it struggles to differentiate RA tasks from parity and RAR. Even though parity and RAR cDFAs share structural similarities with RA cDFAs, this lack of separation is concerning as there is a semantic difference between these task classes–RA cDFAs have rejecting states but parity and RAR cDFAs do not. However, it still successfully clusters cDFAs that are 2-conjunction collapsed and 1-step advanced while also isolating accepting ones in the green region shown in Figure 11d, illustrating the robustness provided by pre-training on RAD cDFAs.

Another point to note is the scales of the Euclidean distance heat maps given in Figure 11c and Figure 11f. The distance scale for RGCN is from 0 to 1.5 whereas the scale for GATv2 is from 0 to 3. Moreover, we see darker colors in Figure 11c, suggesting that GATv2 captures semantic nuances by effectively leveraging the embedding space for robust representations of cDFAs. The analysis results suggest that GATv2 learns a richer representation of the cDFA embedding space, likely due to its attention mechanism, which enables it to attend to relevant features and relationships within the data.

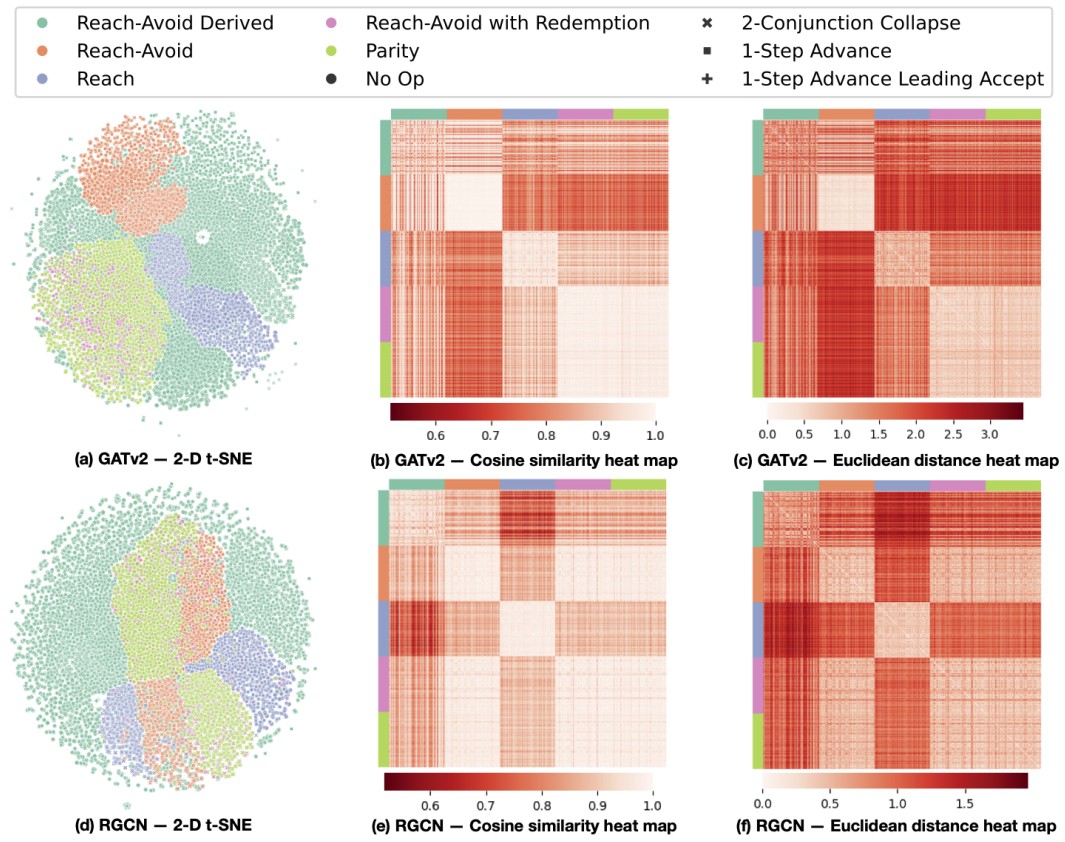

Figure 11: Visualizations of the embeddings generated by GATv2 and RGCN pre-trained on RAD cD-FAs, showing that the learned embedding space reflects the similarities between different task classes.

**C.4 Generalization of policies: pre-trained vs not pre-trained.** Figure 12 compares the generalization strengths of policies with not-pre-trained, pre-trained-and-frozen, and pre-trained GATv2 cDFA encoders. It shows a similar pattern to the learning curves given in Figure 5. Specifically, we see that the frozen one outperforms others while the pre-trained one performs comparably. However, we see that not-pre-trained one does not come close to the others, indicating that learning embedding spaces through pre-training is essential for good performance and generalization to different task classes. Figure 13 presents the number of steps taken in Letterworld to solve the task for the same experiment. We see a similar trend where without pre-training model needs to take a lot of steps to end the episode (either successfully or unsuccessfully). Both pre-trained models take significantly fewer steps while the frozen one solves the task fastest. Combining both figures, we see that the frozen version generalizes well while also learning to solve tasks faster.

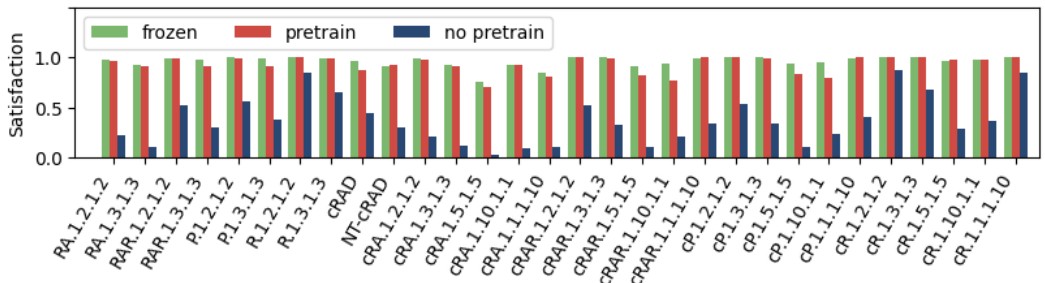

Figure 12: Effect of pre-training on satisfaction generalization capabilities in Letterworld.

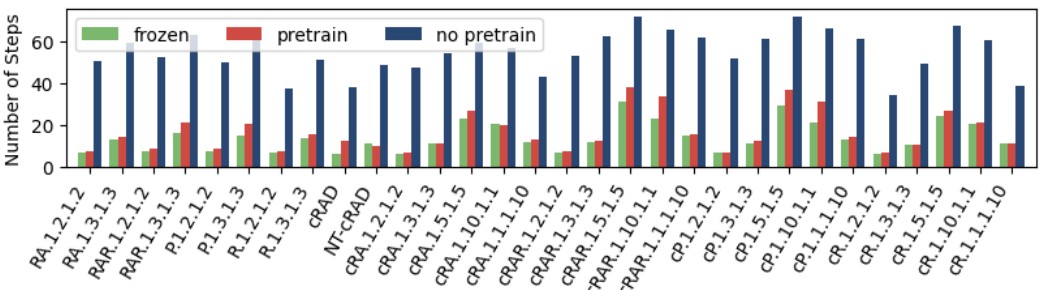

Figure 13: Effect of pre-training on number of steps generalization capabilities in Letterworld.

**C.5 Training curves and steps generalization for other task classes.** We first present the training curves for RA.1.1.1.5 and RAR.1.2.1.2 using RAD pre-trained cDFA encoders. Figure 14 shows that policies trained on RAD and RAR.1.2.1.2 converge relatively faster than the one trained on RA.1.1.1.5. Notice that RA.1.1.1.5 samples cDFAs with more than 3 DFAs with high probability (since the number of conjunctions sampled uniformly at random from the given interval) whereas the probability of sampling such big compositions is lower in RAD (as it sampled from a geometric distribution). Thus, the figure indicates that it is relatively harder to learn big conjunctions.

In Figure 7, we shared the satisfaction generalization results for these policies, testing how well these policies trained on specialized task distributions generalize to other task distributions. Here, we provide the number of steps taken to finish the episode during the generalization experiments in Figure 15. Observe that training the policy on RAD cDFAs results in shorter episodes. However, it is still worth noting that other policies perform comparably and provide good generalization on unseen tasks.

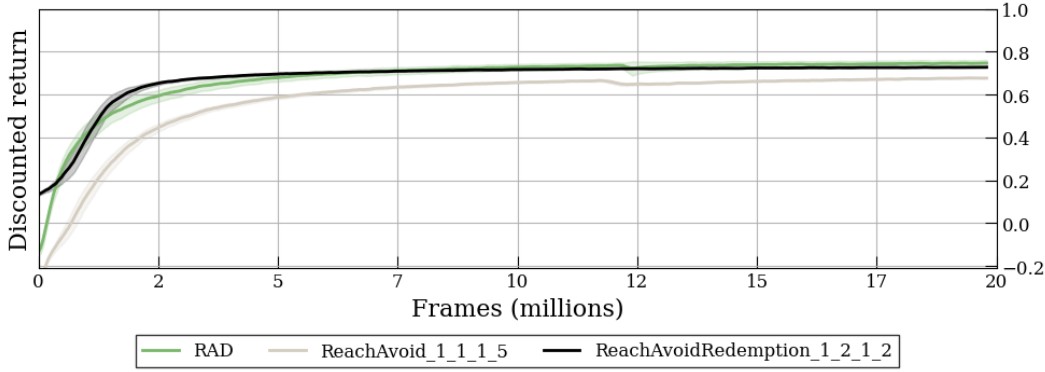

Figure 14: Training curves for RA and RAR policies with RAD-pre-trained GATv2 in Letterworld.

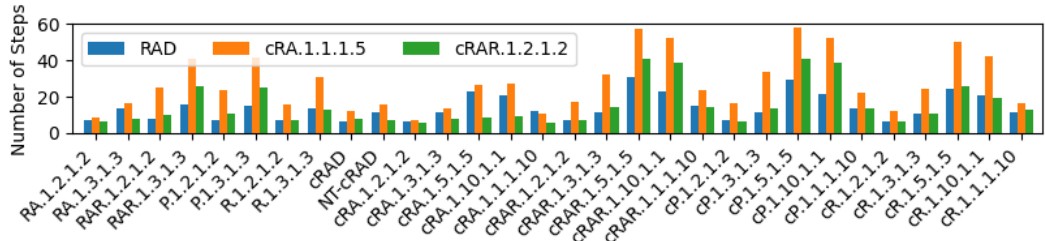

Figure 15: Number of steps generalization comparison between policies trained on RAD, RA.1.1.1.5 and RAR.1.2.1.2.

**C.6 Generalization of pre-trained policies: GATv2 vs RGCN.** Figure 16 present the results. It compares policies with frozen and pre-trained GNNs (both GATv2 and RGCN). In line with the training curves, we can see that GATv2 outperforms RGCN across the board, with the most significant differences on harder task classes with many symbols or many conjunctions. Figure 17 shows the number of steps taken in the same experiments, indicating that GATv2 also learns to complete the tasks with fewer steps compared to RGCN. The results highlight the superior performance of GATv2.

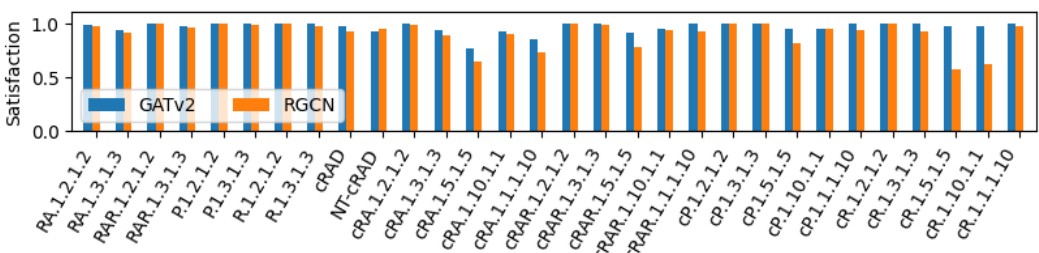

Figure 16: Satisfaction generalization capabilities of policies trained on RAD cDFAs with frozen and pre-trained cDFA encoders in Letterworld, comparing GATv2 and RGCN.

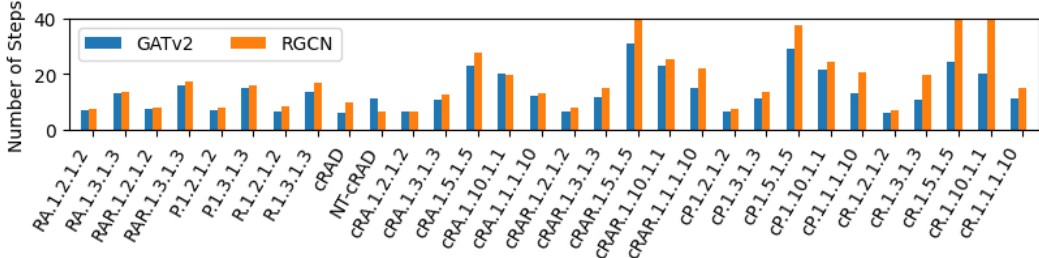

Figure 17: Number of steps generalization capabilities of policies trained on RAD cDFAs with frozen and pre-trained cDFA encoders in Letterworld, comparing GATv2 and RGCN.

**C.7  Generalization of pre-trained policies in continuous domains.** Figure 18 shows the generalization results for a policy (with a frozen and pre-trained GATv2) trained on RAD DFAs in Zones. The results show that our method can generalize to cDFAs with up to 10 DFAs in the composition as long as each DFA's task length is relatively short. We see that it does not perform as well in tasks with length 5 and more. This is probably due to the fact that control and navigation are much harder in continuous domains compared to discrete ones. Therefore, the low accuracies we are seeing for tasks with more length might be due to timeouts. It is worth noting that the policies can generalize up to 10 conjunctions in a cDFA, which is completely out of distribution for the policy. Overall, the results show that training on RAD DFAs provide zero-shot generalization in continuous domains as well.

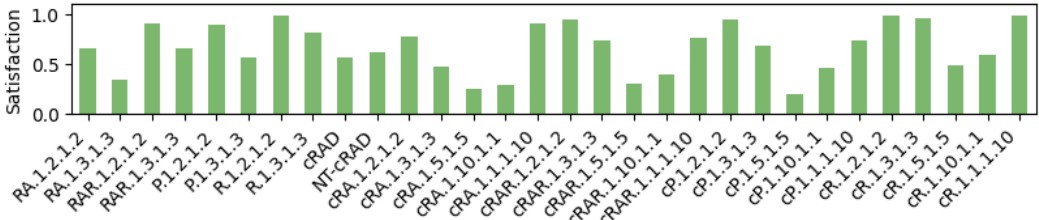

Figure 18: Satisfaction generalization results of policies trained on RAD cDFAs with frozen and pre-trained GATv2 cDFA encoders in the Zones environment (continuous domain).

Figure 19 shows the number of steps taken to complete the episodes in the same experiment. We see that when the policy satisfies a task class with a high probability, it can also complete it with fewer steps. On the other hand, in almost all cases where it could not satisfy the task with high probability, it also took more steps in those episodes, suggesting that the policy might have timed out.

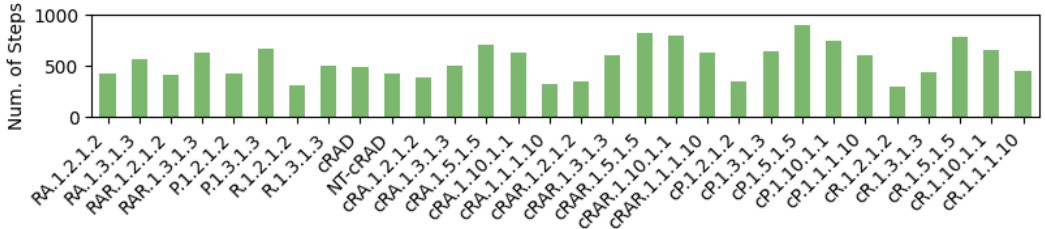

Figure 19: Number of steps generalization results of policies trained on RAD cDFAs with frozen and pre-trained GATv2 cDFA encoders in the Zones environment (continuous domain).

**C.8 Generalization of pre-trained policies: RAD cDFAs vs LTL2Action.** To compare our approach to the closest work in the literature, we design an experiment with LTL2Action [38]. Recall that one of our main contributions is our RAD pre-training procedure that provides generalization across different tasks. Defining such a general LTL task class is not a trivial problem and is a contribution on this own. Moreover, the intuitions used to develop the RAD cDFA class are not applicable to LTL as it is not as easy to generate interesting and satisfiable LTL formulas. Therefore, for LTL2Action, we define a *joint* task distribution consisting of partially ordered and avoidance tasks, which are task classes defined in [38]. To sample LTL tasks, we flip a coin and pick a samplers.

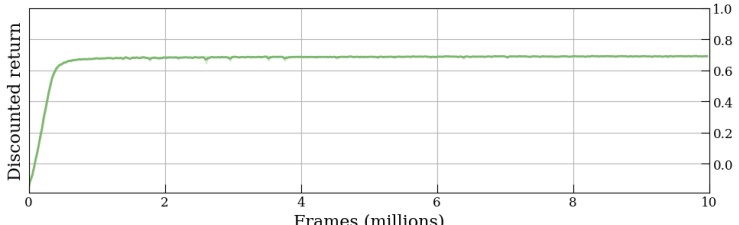

Figure 20: Learning curve for pre-training LTL policies on the joint distribution of PO and AV.

Figure 20 present LTL2Action's learning curve for the pre-training of an RGCN in the joint LTL task distribution. As the figure shows, the model can easily learn to distinguish these LTL tasks.

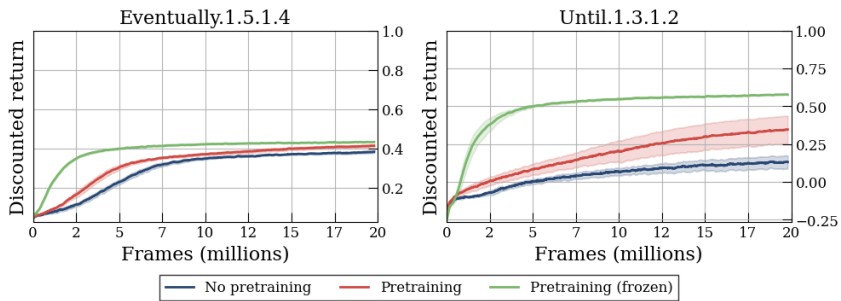

Figure 21: Learning curves for training LTL policies on PO and AV LTL tasks.

We then take the pre-trained RGCN and use it to train a policy that is exclusively trained on one of these task classes, i.e., partially ordered or avoidance LTL tasks. Figure 21 presents the learning curves for this training. The figure shows that if we let the RGCN get gradient updates during training, then pre-training does not help as much as freezing the network. We saw a similar behavior for RGCN in Figure 5, which further justifies our choice of using a network with an attention mechanism. Furthermore, in [38], authors show that pre-training on specific task classes helps training even when we let the RGCN continue getting gradient updates. Combining these results from [38] with what we are seeing on Figure 21 suggests that when pre-trained on rich classes with multiple distinct task patterns, RGCN need to *forget* about the other task classes it learned during pre-training and has to focus its capacity on the specific task class it is being trained on. We do not see such a gap in the learning curves when we let GATv2 get gradient updates, see Figure 5, suggesting that GATv2 does not suffer from the same issue as RGCN, and it learns more robust cDFA embeddings.

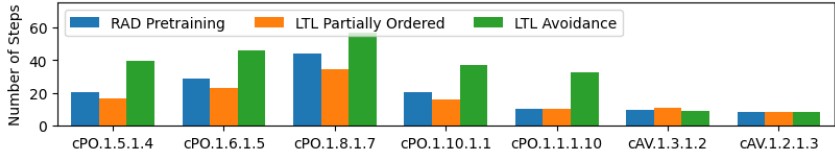

Figure 22: Number of steps generalization capabilities of LTL policies vs RAD-cDFAs policies.

Figure 22 presents the number of steps taken for the generalization experiments, showing that in all cases our policy trained on RAD cDFAs perform comparably to the specialized LTL policies.

# D Hyperparameters and Compute Usage

Both for the RGCN and GATv2 cDFA encoders, we use a hidden dimension size of 32 and perform 8 message passing steps. Also, for GATv2, we use 4 heads for the multi-head attention.

For both Letterworld and Zones, we implemented the same actor and critic architectures. The actor network comprises three fully connected layers, each with 64 units and ReLU activations, while the critic network has three fully connected layers with 64, 64, and 1 units, using a Tanh activation. For pretraining, we simplified the actor and critic to single-layer models without hidden units, allowing the cDFA encoder to learn a rich embedding space. For environments with discrete actions, the actor's output was passed through a log softmax and a logit layer. In continuous action settings, we used a Gaussian distribution, parameterizing the mean and standard deviation with two separate linear layers from the actor's output. To construct observation embeddings, for Letterworld, we employed a 3-layer convolutional network with 16, 32, and 64 channels, 2×2 kernels, and a stride of 1. In Zones, a 2-layer fully connected network with 128 units per layer and ReLU activations was used.

Table 1 shows the hyperparameters used for every training run in the experiments section. Each seed in the experiments section was run as an individual Slurm job with access to 4 cores of an AMD EPYC 7763 running at 2.45GHz and access to at most 20gb of memory. RAD pre-training experiments took approximately 14 hours each for 10 million timesteps, policy training in Letterworld took approximately 55 hours (50 hours for the frozen GNN version) for 20 million timesteps, and policy training in Zones took approximately 120 hours (70 hours for the frozen GNN version).

| Hyperparameter | Pretraining Env. | Letterworld Env. | Zones Env. |
|---|---|---|---|
| Learning rate | 0.001 | 0.0003 | 0.0003 |
| Batch size | 1024 | 32 | 2048 |
| Number of epochs | 2 | 4 | 10 |
| Discount | 0.9 | 0.94 | 0.998 |
| Entropy Coefficient | 0.01 | 0.01 | 0.003 |
| GAE($\lambda$) | 0.5 | 0.95 | 0.95 |
| Clipping $\epsilon$ | 0.1 | 0.2 | 0.2 |
| RMSprop $\alpha$ | 0.99 | 0.99 | 0.99 |
| Max. grad. norm. | 0.5 | 0.5 | 0.5 |
| Value loss coef. | 0.5 | 0.5 | 0.5 |

Table 1: PPO pre-training and policy training hyperparameters.

