# OpenReview forum: "Compositional Automata Embeddings for Goal-Conditioned Reinforcement Learning"
_NeurIPS.cc/2024/Conference — NeurIPS 2024 poster_

### Official Review · Reviewer_7HER · 2024-07-11

**Soundness:** 2
**Presentation:** 2
**Contribution:** 2
**Rating:** 5
**Confidence:** 4

**Summary:**

This paper tackles Goal-conditioned Reinforcement Learning (RL) when used with Temporal Logic Objectives. The benefits of directly considering Deterministic Finite Automatons (DFAs) as the task definition leads to a new class of Compositional DFAs introduced that cover a conjunction of several tasks. These cDFAs can be encoded using a Graphical Neural Network that is pretrained on a novel task distribution (RAD). Experimental results show the approach generalizes to several task classes.

**Strengths:**

- Compositional DFAs are a more general form of multi-task descriptions. The proposed method is a straightforward way to encode these objectives and solve them.
- The range of task classes considered is extensive and covers a variety of problem settings.

**Weaknesses:**

- The approach follows the similar structure as LTL2Action [41] but replaces the LTL Module with a different GNN (GATv2) and considers cDFAs or compositions/conjunctions of DFAs. The modifications w.r.t. LTL2Action apart from this and the pre-training step over a new randomly sampled (RAD) task distribution are novel yet arguably incremental.
- Baselines appear lacking which bring into question the merits of the approach. A direct comparison to LTL2Action [41] and GCRL-LTL [32] using the monolithic DFA of the cDFA in the experiment could be possible and if not, the reason should be justified.
- The paper is at times hard to follow and overloaded with notations and conventions. Some improvements could be made in the description of RAD pre-training (see questions) and the task nomenclature 1.X.1.X in the experiments.

**Questions:**

1. Should the statement in L35 be better quantified or validated with references? "DFAs represent all temporal tasks that become Markovian by augmenting the state-space
with finite memory”. Does this hold when Signal Temporal Logic (STL) is considered a temporal task language as well?
2. Is Reference [7] a concurrent submission or a typo?
3. In Figure 5, what are the black elements in the Legend? Why are they not present in the figures?
4. What do the envelope colors in Figure 2 indicate? Does the encoding generalize to conjunctions of the predicates on edges (e.g., Red & Green)? Are these considered during pretraining?
5. Defn. 4.1. could use some clarity. How is a random transition defined in a mutation? Is this simply changing the target node of a given transition to a random state in the DFA? What is being “minimized" here?
6. Minor grammatical error - Should DFA and DFA(s) be used interchangeably? I would think the following references to DFA should be changed to DFAs (L31, L33, L80, L85-87, and more)

**Limitations:**

The limits of cDFA encoding are not fully explored (mentioned).

---

> ### Author Rebuttal · Authors · 2024-08-06
>
> _Thank you for the time and effort put into your review._
>
> ### Comparison to LTL2Action and GCRL-LTL.
> We provided an in-depth empirical comparison along with a detailed discussion with LTL2Action in Appendix C.8. We plan to use the extra space afforded by a camera ready to move it into the final version of the paper.
>
> ### Comparison to GCRL-LTL
> In the related work section of the paper (lines 94-100), we provided a discussion of GCRL-LTL which justifies why we did not compare against that method. Please see the **Baselines** section in the meta-rebuttal for a more detailed discussion. We will clarify this in the final version of the paper.
>
> ### Clarity, presentation, and notation
> Thank you for the feedback.
> We will make the necessary fixes pointed out by the reviewer and further simplify the presentation in the final version.
>
> ### DFAs represent all temporal tasks that become Markovian by augmenting the state-space with finite memory
> Thank you for bringing this up. We will clarify what this means in the final version
> as it is indeed confusing / ill-posed.
> Here, a task is taken to be a Boolean property of an episode: you either perform it or you don't.
> By making a task Markovian we mean that the state of satisfaction can be determined
> by just looking at the current state, e.g., did you reach a goal state?
> Importantly, we are assuming that each state is labeled from a finite set.
> Mechanically, one can then identify the DFA task from the transitions of
> the finite states since the properties are only defined over the labels of the finite states.
>
> ### Signal Temporal Logic (STL), finite memory, and DFAs
> The short answer with STL is no. STL is not generally representable with finite memory.
> A longer answer with STL is that it depends on the specific STL formula and the semantics applied to (i) time, (ii) the signals, and (iii) how finite truncations of the signals are handled.
> But, per the earlier point, if the STL formula has a finite memory monitor,
> then yes it can be encoded as a DFA.
> In such cases, the DFA often corresponds to syntactic progression as seen in LTL2Action.
>
> ### Is Reference [7] a concurrent submission or a typo?"
> Reference [7] is neither a concurrent submission nor a typo. It is a non-archival workshop paper that we have written. It successfully solves the reward sparsity problem with DFA rewards in off-policy RL algorithms using hindsight experience replay.
> ### In Figure 5, what are the black elements in the Legend? Why are they not present in the figures?
> Zooming into that figure will reveal that each point has a distinct shape representing the kind of operation applied to the sampled cDFAs.
> These shapes correspond to the black elements in the legend. We will make this clearer in the final version.
>
> ### What do the envelope colors in Figure 2 indicate?
>
> Those colors are just there to help track where the messages came from. We will make this clearer in the final version.
>
> ### Does the encoding generalize to conjunctions of the predicates on edges (e.g., Red & Green)? Are these considered during pretraining?"
> Our work does not treat symbolic relations on the edges, however we think it is an interesting
> direction. Conjunctions may potentionally handled with a many hot encoding, but this is
> purely speculation.
>
> ### Defn. 4.1. DFA Mutation
> Consider the corresponding adjacency matrix of a DFA, which is a 0-1 matrix indicating the transitions between states. We randomly sample an entry in this matrix and toggle that entry (if the entry is 0, make it 1; and if the entry is 1, make it 0). This is precisely what a mutation is. We will update the definition in the final version and make it clear.
> ### "What is being “minimized” here?"
> The DFA is being minimized. Specifically, the number of states is being minimized.
> Conceptually, this corresponds to merging states that are interchangable (have equivilent residual languages).
> This results in a canonical representation that is irreducable.
> See https://en.wikipedia.org/wiki/DFA_minimization for details.
>
> ### Should DFA and DFA(s) be used interchangeably?"
> Thank you for pointing out this inconsistency. They are indeed not interchangable
> and we will correct these abbreviations in the final version.

---

> > ### Comment · Reviewer_7HER · 2024-08-13
> >
> > I thank the authors for the clarifications. The main contributions are clear (RAD pretraining/embeddings) and I understand how DFAs can encode many tasks captured by different temporal logics. I will raise my score accordingly since I beilieve the method presents a useful technique to generalize over logical tasks.
> >
> > Rather than a qualitative argument as to why other related approaches (GCRL-LTL) are different, I would have been interested to see simulations on a sample task/specification. Further, I believe the paper could use some presentation refinements (as suggested) and look forward to seeing the final version.

---

> > > ### Author Response · Authors · 2024-08-14
> > >
> > > _We thank the reviewer for their response and for raising their score._
> > >
> > > We will make sure to include all of the suggested presentation refinements into the final version and believe it will significantly help the clarity of the paper. Space and time permitting, we plan on running empirical comparisons against the related approaches (GCRL-LTL) on the types of problems described in the meta-rebuttal to backup the qualitative argument.

---

### Official Review · Reviewer_bN73 · 2024-07-13

**Soundness:** 4
**Presentation:** 4
**Contribution:** 4
**Rating:** 8
**Confidence:** 4

**Summary:**

This paper extends the framework of goal-conditioned reinforcement learning to support temporally-extended goals specified as (compositions of) deterministic finite automata (cDFA), avoiding the limitations of state-based goals and the ambiguities of natural language specifications. The authors introduce a graph neural network architecture for embedding cDFAs to vectors via message passing. They then develop a pretraining distribution for learning cDFA embeddings, which is constructed by leveraging the insight that paths in a DFA correspond to a series of reach-avoid tasks, motivating a distribution of cDFAs that are derived from sequential reach-avoid DFAs. In experiments, the authors show that pretraining on this distribution produces generalizable embeddings that capture the structure of DFA space, while also enabling the training of generalizable cDFA-conditioned policies. The authors run these experiments on two labeled MDPs, Letterworld and Zones, showing that embeddings or policies trained on only RAD DFAs or other DFA subsets often achieve high goal satisfaction rates even when presented with cDFAs with longer task lengths and higher composition numbers. cDFA conditioned policies using a frozen cDFA embedder trained on the RAD distribution also lead to high satisfaction rates across the board in Letterworld, with reasonable strong performance in Zones (a harder continuous MDP).

**Strengths:**

This is a very well-written and thorough paper that made for a pleasant and insightful read. Despite not working much with DFAs or GNNs, I found both the overall idea and the low-level details very easy to understand, because all key concepts are carefully but concisely explained, with figures that clearly illustrate what a cDFA is, how it can fed into a GNN, and what the structure of SRAs and RADs look like. The design of the RAD pretraining distribution was also very insightful, demonstrating careful thinking about what motifs ought to be repeated across a large class of DFAs (namely, reach-avoid tasks), and how that could be exploited to enable generalization. Experiments were very clearly laid out and thoroughly executed, covering a wide range of DFA classes to test generalizability, and demonstrating that the proposed method actually does lead to cDFA-conditioned policies that generalize quite well across the board.

In terms of motivation and impact, I think it's great that people are exploring more expressive and well-defined notions of goals in the context of goal-conditioned RL. While working with formal specifications is not very popular in machine learning these days, I think a strong paper like this one helps to elevate the profile of using such specifications to enable more interpretable and generalizable RL systems. As such, I think this paper could have a decently large impact within the fields of both goal-conditioned RL and people working on formal methods in RL / ML.

**Weaknesses:**

As the authors themselves note, probably the main weakness / limitation of work that relies on symbolic specifications like cDFAs or LTL formulae is that it requires MDPs with a labeling function. This limitation isn't insurmountable --- for e.g., it should be possible to use computer vision algorithms to segment and label states of the world from pixels -- but in the final version of the paper, I think it'd be good to discuss this a bit more, since it might help readers appreciate how work like this could be applicable without a oracle labeling function.

Another slight weakness of the paper is that most of the experiments focus on Letterworld, with weaker average performance in the more realistic environment of  Zones. This raises questions about how applicable the method will be to more realistic tasks. With the extra page in the final version, I think it'd be good to show images of both environments, and examples of harder cDFAs in those environments, just to illustrate how difficult it might be to solve those tasks. Otherwise it might be easy to write-off e.g. Letterworld as a toy domain even though with cDFAs the tasks can be quite long-horizon. It might also be good to say a bit bout how performance could be improved for harder tasks like Zones.

Finally, I think one thing that bears discussing in the final version is how this framework might enable safety/correctness of the policy, not just interpretability of the specification. While it's no doubt a safety benefit to have more interpretable specifications that say exactly what correct behavior is, this on its own doesn't guarantee the resulting learned policies are safe. I wonder if the authors could say a bit more on this point, perhaps by connecting to work on formal verification of NNs, or by combining cDFA-conditioned RL with constrained planning methods, using the learned policy to accelerate provably safe planning.

**Questions:**

There were a couple of minor things I didn't fully grasp, which it'd be great to clarify:

- When constructing/sampling a SRA DFA (e.g. RHS of Figure 3) how are reach or avoid transitions created? Is the reach transition always a transition to the accepting state? Or is it to a a state that eventually leads to an accepting state? Is it the case that there is a single long path to the accepting state? And is the avoid transition always to a single sink state? I couldn't fully understand this from Appendix B.

- In Figure 4, in the panel for Zones, where is the (non-frozen) GATv2 pretraining line baseline? And am I right to say that the RGCN baseline was not trained for Zones?

- For each MDP with its own labeling function and alphabet, do you basically to do the RAD pretraining phase for specifically that alphabet of symbols? Or is there hope of training a cDFA embedder that generalizes across different MDPs with different labeling functions and alphabets?

**Limitations:**

The authors have adequately discussed a number of limitations of their work. It would be good to also discuss the fact that interpretable specifications (in the form of cDFAs) do not immediately lead to verifiably safe behavior.

---

> ### Author Rebuttal · Authors · 2024-08-06
>
> _Thank you for the time and effort put into your review._
>
> ### Regarding the comment on the labeling function.
> We agree with the potential usage of computer vision algorithms to segment and label states of the world from pixels. We think there is a lot of interesting and fruitful future work that could address this limitation. We will use the additional space to comment on this issue and hint at possible directions for future work in the final version of the paper.
>
> ### Weaker performance in Zones.
> The performance difference between the Letterworld and the Zones environments, as the reviewer pointed out, is due to the fundamental difficulty that comes with continuous control. That is, in continuous domains, it is harder to learn control from an uncountable domain of inputs, and it takes longer to reach a goal, resulting in lower discounted return. As suggested, we will include images from both environments to highlight these points and the fundamental difficulty of continuous domains along with a discussion on how to mitigate these difficulties.
>
> ### Safety/correctness guarantees
> Thank you for the question. At this point, without further research, we believe it is hard to say anything precise about the safety of the learned policies. To the best of our knowledge, there are still scalability issues with the state-of-the-art NN verification literature; however, the constrained planning methods direction pointed out by the reviewer is definitely interesting and is part of our broader research agenda. Another [promising approach is also have the policy output a correct certificate, such as a reach-avoid martingale, to make the verification problem simpler](https://ojs.aaai.org/index.php/AAAI/article/view/26407).
>
> ### How are reach or avoid transitions created
> Conceptually, the reach/avoid transitions are indirectly constructed by first starting with a chain of N states and a failure state, FAIL:
>
> > $S_1 \rightarrow S_2 \rightarrow \cdots \rightarrow S_N$
>
> For each state $1,\ldots,N$, the tokens are partitioned into reach/avoid/noop.  The reach tokens advance that state ($S_i \rightarrow S_{i+1}$). The avoid tokens transition to FAIL. RAD is derived by mutating the transitions (toggling entries in the transition matrix) and minimizing. Conceptually, this acts to reintroduce features like cycles, skip connections that offer more than a single path, while avoiding fully connected ``random'' graphs.
>
> ### Question about Fig. 4.
> Due to time and compute.
> Running experiments in the Zones environment requires much more compute resources and time than Letterworld; therefore, we only run experiments for the best (frozen and pretrained GATv2) and baseline (GATv2 with no pretraining) configurations of the framework in the Zones environment. For the same resource reason, in Zones, we did not train the policies using embeddings produced by an RGCN model. We believe that the Letterworld experiments give enough evidence to conclude that embeddings of a pretrained and frozen GATv2 give the best performance in the continuous domain, but can include more experiments if it would be useful.
>
> ### Question on training a general cDFA embedder
> Thank you for the question. To answer this, first note that the pretraining is independent of the downstream application / MDP. Second, without environment semantics, the tokens are just labels that can be permuted, relabeled, etc. Thus, changing the labeling is as simple as deciding the index of each token. Therefore, one can pretrain a GATv2 model on RAD cDFAs with N symbols, and then use it in any environment and downstream application with up to N symbols.
>
> Of course, in order to use the embedding of the environment, one must somehow ground/order the tokens. In our experiments, this is implicit through the evolution of the embeddings (as the DFA progresses, we provide the corresponding embedding). In the future, one could imagine sending as side information an embedding of each token, but this is purely speculative. We will make it clear in the final version.

---

> > ### Comment · Reviewer_bN73 · 2024-08-13
> > **Thank you for the response.**
> >
> > Thank you for the answers to my questions. I continue to think this is a strong paper, and will be maintaining my score. Having read the other reviews, I also think it would be valuable to make the comparison with LTL2Action more explicit and move it into the main text.

---

> > > ### Author Response · Authors · 2024-08-14
> > >
> > > _We thank the reviewer for their response and for arguing that this is a strong paper._
> > >
> > > We will certainly move the comparison to LTL2Action into the main text in the final version of the paper to make sure that it is clear to readers.

---

### Official Review · Reviewer_Ww4e · 2024-07-15

**Soundness:** 3
**Presentation:** 3
**Contribution:** 2
**Rating:** 6
**Confidence:** 4

**Summary:**

This work focuses on goal-conditioning policies using DFAs. This leverages the ability of DFAs to be composed - in this case the work focuses on conjunction. Two main pieces are introduced in leverging DFAs: 1) a GATv2 model which provides task embeddings from the DFA, 2) pre-training of the GATv2 on Reach Avoid Derived compositional automata which trains the embeddings with a basis representation in terms of the RAD embeddings which can then be used to represent other DFA embeddings. Once the  DFA embeddings is obtained it is appended to the environment state space and used to train a policy. Experiments demonstrate the benefit of GATv2 over an alternative RGCN (when pre-training and not freezing the embeddings) and also demonstrate the benefit of pre-training the task embeddings, particularly when freezing the embeddings after.

**Strengths:**

# Originality
The use of DFAs and similar representations of tasks is a popular idea in RL at the moment. However, this work focuses on the use of the DFA within a longer pipeline to obtain the embeddings. This use appears novel as most works I am aware of tend to use fairly literal and inflexible representations of the DFAs or state machines. This work then also puts some interesting pieces together - DFAs for task representations, GATv2 to represent a graph, RAD pretraining. Overall this all supports the novelty of this work and I do believe this is a strength.

# Quality
The aim and hypothesis of this work is clear. Each step of obtaining the task embedding is justified and validated with experiments. The ablation study does demonstrate the utility of both GATv2 and the pretraining from RAD tasks. Moreover the additional experiments and visualisations support the fact that GATv2 is learning to produce reusable and semantically meaningful task embeddings.

# Clarity
Overall the paper is well written, notation is used sparingly and intuitive. This is particularly useful as the combination of DFAs and MDPs can often become very notation heavy. I appreciate the effort placed in making this work clear. Images and visualization are also neat and clear and support understanding.

# Significance
Enabling generalization in RL, and specifically task generalization is an important problem. It has particular utility for enabling safe and adaptable RL agents. This work provides a new perspective and step towards this goal. I do think that this work would lead to future work and new ideas.

**Weaknesses:**

# Clarity
Figure captions could elaborate more. As it stands they are vague and do not add any explanation or support understanding of the figure. Secondly, Section 3.1 is unclear due to the omission of an explanation for the DFA modifications. As a result it is unclear why any of these steps are needed, but this seems to be an important and early part of the pipeline. Thus, it leaves the entire pipeline feeling unclear or at least on an uneasy grounding.

# Significance
Unfortunately, I think the significance of this work may be its biggest weakness. Firstly, it is not clear to me that it is trivial to extend the findings on conjunctions will generalize to general boolean combinations as is mentioned in lines 144 and 145. Secondly, the curse of dimensionality which results from conjunctions of DFAs is never addressed or touched on. Perhaps the vector representation of task embedding does support generalization - and the visualizations would even support this - but it is not discussed. This is even more worrying when considering that a fixed dimension for the task embeddings is used. Thirdly, the experiments do support the answering of the research questions posed. However, the experiments do not compare to any other baselines from the literature, which makes it difficult to contextualize and build from this work. Finally, and I acknowledge that the authors point this out in the limitations section and so I will take this point lightly, but there is no insight into how the task embeddings support generalization. This does unfortunately limit the significance of the work.

**Questions:**

1. How would the results of the generalized agents with task embeddings compare to specialized agents?
2. In Figure 4, why do the Zones results only have two models?
3. At a high level, how would you extend the approach to incorporate general boolean combinations

If the authors could address my concerns with regard to the curse of dimensionality, and answer these questions I would likely advocate for acceptance.

**Limitations:**

Limitations are mentioned and I do not see any omissions.

---

> ### Author Rebuttal · Authors · 2024-08-06
>
> _Thank you for the time and effort put into your review._
>
> ### Regarding Clarity of Sec. 3.1.
>
> We appreciate this feedback on clarity so we will include an explanation for the specific graph encoding we use. Essentially, we apply four operations on a DFA to construct a graph encoding:
>
> 1. Add intermediate nodes between states to encode edge constraint information. **Explanation:** We want edge constraints to be used during message-passing. By introducing these intermediate nodes with edge constraint information, we make sure that each node receives messages including edge constraints needed to get to the next states.
> 2. Reverse all the edges. **Explanation:** We want each state to receive information from future states since the goal is to find an accepting path in the DFA.
> 3. Add self-loops. **Explanation:** We want each node to receive messages from itself to avoid forgetting the local node message. It is a common practice in the GNN literature.
> 4. Add edges from the nodes corresponding to the initial states of DFAs to the “AND” node. **Explanation:** The "AND" node can be understood as a conjunction operator aggregating messages from each DFA in the composition. It is connected to the initial states of each DFA because we want this operator to receive information from the current state of each DFA.
>
> ### Curse of Dimensionality.
> By the curse of dimensionality, if the reviewer means the exponential blow-up caused by taking the conjunctive composition of DFAs, please see the cDFA section of the meta rebuttal which clarifies how we avoid such a blow-up.
>
> ### Comparison with other baselines
> Please see the baselines section of the meta-rebuttal.
> To directly address your question,  we will move the experiments comparing our work with LTL2Action (given in Appendix C.8) to the main body of the paper to help further contextualize the proposed method.
> The key result is that our approach does indeed generalize better given the RAD pretraining.
>
> ### Insights into how the task embeddings support generalization.
> We definitely believe that further research is crucial to understand why and how the task embeddings support generalization. However, we believe this is a research question on its own and should be investigated in future work. Please also note that our embedding space analysis given in Fig. 5 in the main body and Fig. 9 in the appendix provide an intuition for the generalization results we see in the paper by visualizing the structured nature of the learned embedding spaces.
>
> ### How would the results of the generalized agents with task embeddings compare to specialized agents?
> In our experiments, we did not see any significant difference between RAD pretrained agents versus agents trained for a single task class when evaluated on the same task class; therefore, we omitted these results. However, we can include such a comparison in the final version of the paper.
>
> ### In Figure 4, why do the Zones results only have two models?
> Due to time and compute.
> Running experiments in the Zones environment requires much more compute resources and time than Letterworld; therefore, we only run experiments for the best (frozen and pretrained GATv2) and baseline (GATv2 with no pretraining) configurations of the framework in the Zones environment. For the same resource reason, in Zones, we did not train the policies using embeddings produced by an RGCN model. We believe that the Letterworld experiments give enough evidence to conclude that embeddings of a pretrained and frozen GATv2 give the best performance in the continuous domain, but can include more experiments if it would be useful.
>
> ### At a high level, how would you extend the approach to incorporate general boolean combinations?
> Please see the meta-rebuttal section on cDFA.

---

> > ### Comment · Reviewer_Ww4e · 2024-08-12
> > **Comment by Reviewer Ww4e**
> >
> > I thank the authors for their thorough response.
> >
> > * For the additions to improve clarity of Section 3.1. These additions are what I had in mind and will address my concern.
> > * For the concern on the curse of dimensionality: I note that the cDFA means the exponential blowup in states does not explicitly occur. However, the vector representation of the task space must still be able to disambiguate the composed tasks from all other tasks. This is my main concern, that the fixed vector size will either need to be excessively large to begin with or will rapidly become exhausted. This is an empirical question of how quickly this vector space representation will become difficult to use due to the exponential increase in task space from composition. The cDFA point above does not seem to address this, unless I am missing something.
> > * A comparison to LTL2Action seems useful and to my knowledge is the most appropriate baseline. This would correct the original omission in my opinion.
> > * I agree that investigating the task embedding space would support future work. However, it is unfortunately related to my above point that it is unclear how quickly the empirical results of this work would decay without such an understanding. To me this isn't a difference between acceptance or rejection but does limit the significance of the work at the upper end. Particularly, because many benchmarks which do not use task embeddings are guaranteed to generalise arbitrarily. The task embedding having a theoretical limit in its representation is a weakness of the proposed method which is very difficult to assess - I admit this - but is still a limitation none the less. However, I acknowledge that this is pointed out by the authors and so I emphasise that I am not using this to decide on acceptance.
> > * I think this concession on the zones experiment needs to be stated explicitly. This could also be of importance in assessing the scaling capacity of the proposed method. However, once again I can accept these concerns being left to future work as it is stated in the limitations section.
> > * I acknowledge that I have read the reviewers comments in the general rebuttals.
> >
> > Overall I think that if the changes to clarity are made as proposed then I would be comfortable seeing the work accepted. Unless I am missing something, the score is limited due to the difficulty of assessing the scalability of the model which is a weakness unique to using task embeddings compared to approaches which do not such as those cited by Reviewer A6oy. This needs to be stated too. However, in acknowledgement of the proposed changes and some helpful clarification I will raise my score to a 6. However, I think the discussion with the other reviewers appear useful and given their in-depth responses I would defer the outcome to be weighted towards their reviews. Thus, I will be lowering my confidence.

---

> > > ### Author Response · Authors · 2024-08-12
> > >
> > > _We thank the reviewer for their response and for raising their score._
> > >
> > > We are glad to hear that you agree that the comparison to LTL2Action is the most appropriate baseline and that the additional clarity in Section 3.1 addresses your concerns. We agree that further investigation into the task embedding space to understand its scalability (in more complex environments such as Zones and with larger tasks) is an exciting direction of future research. We will make sure to address these limitations and future directions clearly in the final version of the paper.
> > >
> > > It is true that the fixed representation size does provide a theoretical limitation on its representation capacity as the size of the task space increases. We have two reasons to be hopeful that this will not pose a major limitation in practice:
> > > 1. This exponential blowup is a theoretical limitation for any sequence modeling task and yet, fixed size embedding have proven to be successful in computer vision (e.g., CLIP) and natural language (e.g., word2vec) domains.
> > > 2. DFAs (compared to unstructured representations such as natural language) enable a natural finite approximation of the embedding space: _planning over the DFA up to a limited depth_. A fixed size embedding space might naturally converge on this approximation.

---

### Official Review · Reviewer_A6oy · 2024-07-24

**Soundness:** 2
**Presentation:** 2
**Contribution:** 2
**Rating:** 5
**Confidence:** 5

**Summary:**

This paper considers the multi-task setting where each task is a temporal logic task specified by a conjunction of deterministic finite automata (cDFA). To address the sample efficiency and generalisation problems present in this setting, they propose a method for generating good cDFA embeddings which can then be used by a universal value function approximator to learn general policies. Precisely, they propose converting a given cDFA into a graph which can then be passed to a graphical neural network (GNN) like GATv2 to create a vector embedding of the task. Similarly to prior work that demonstrated the benefit of pretraining GNN embeddings on LTL tasks (LTL2Action), this paper also proposes the *reach avoid derived* DFA (RAD-DFA) task distribution to pretrain the GNN embeddings. They then demonstrate that using frozen pretrained embeddings leads to better performance than finetuning, and no pretraining leads to abysmal performance. Finally, they also demonstrate that their pretraining procedure leads to reasonable cDFA embeddings and help with generalisation.

**Strengths:**

## Originality
- This work combines three established directions of research, namely: temporal logic task specifications (in the form of DFAs specifically), goal conditioned RL (using UVFAs specifically), and graph neural networks (GATv2 specifically).
- The intersection of these three subfields is of more recent interest and only just emerging in the literature. Thus, there is an element of originality in the specific manner in which they are combined in this work with the aim of improving sample efficiency and generalisation in RL.
- Also the specific manner in which a GNN is pretrained to generate good DFA embeddings (by introducing the RAD pretraining task set) appears to be fairly new.

## Quality and Clarity
- The work provides mathematical formulations for the main concepts needed to understand the approach, which aid the quality of the work.
- Assumptions, while not formally stated are mentioned too.
- The explanation of the framework is also detailed and coherent which makes the contributions of this work more apparent.
- The paper includes a number of Figures which are helpful, clear and in the case of Figure 1, aids the message of the work quite a lot.
- The paper includes a number of experiments that illustrate the goal embeddings learned by the proposed approach, and the resulting sample efficiency and generalisation over specific task distributions.

## Significance
- Temporal logic instruction following is an important direction of work and has wide applications---for example in multitask, lifelong learning and safe RL.
- So leveraging GCRL (with UVFAs) and pretrained GNN embeddings to improve sample efficiency and optimality---important issues in general in RL---for DFA specified tasks can be particularly impactful and widely adopted. In this sense, this work could guide future work and ideas.

**Weaknesses:**

# Major

## Originality
- Overall, this work is very similar to LTL2Action (Vaezipoor et al. 2021) and does not deviate particularly far from the work presented there beyond considering DFA specified tasks instead of LTL ones, which is just a choice of task specification language (regular/DFA vs LTL/BuchiAutomata). Also the paper claims that LTL2Action is limited to "finite" LTL (line 82). It is not clear what is meant by "finite" here. If it means finite trace, then that is incorrect. LTL2Action is applicable to any LTL specification.
- In addition, relying only on a UVFA for sample efficiency and generalisation is a widely known idea that has also been explored in prior works. While the exact implementation here may be new (such as the specific way the goal/task embedding is obtained), the ideas are familiar from a function approximation front.
- The paper introduces the cDFAs, but this does not seem novel since they don't seem meaningfully different from DFAs and the distinction does not seem relevant for the proposed approach.  Hence this just adds more notations and terminologies. Also, the footnote on page 2 says "With negations and disjunctions omitted as straightforward extensions". It is not clear how they are straightforward, unless cDFAs are indeed just DFAs as I mentioned previously.

## Quality and clarity
### Mathematical formulations
- I found it very hard to fully understand the proposed framework and judge the soundness because there are a number of incorrect statements and missing information.
- The MDP definition (Def 2.1) is wrong. An MDP is a five tuple M = (S,A,T,R,$\gamma$) where $R$ is the reward function and  $\gamma$ is the discount factor (optional when $\gamma=1$). It is not a triple M = (S,A,T). The objective of goal-conditioned RL is not to maximize the probability of generating a path containing g (lines 127-128), but instead to maximise the value function (the expected cumulative rewards).
- It is fine if the authors want to focus on the maximisation of the success probability, but then the paper needs to describe how that will be achieved in the RL setting defined.
- The reward function and discount factor for the problem formulation are never defined. The only reward function mentioned in the mathematical formulations is the one used for pretraining the GNN. Since the experiments uses the same reward scheme for learning policies, I suggest the authors formally state that in their problem formulation. Additionally, Table 1 shows that the experiments use a discounting, meaning that the learned policies *do not* correspond to the maximisation of success probability.
- The paper says that goals are defined in the augmented MDPs (line 164), but the DFA-conditioned policy uses the environment states and not the augmented ones (Def 3.1).
 - The paper uses message-passing neural network (MPNN) (line 173) as a core factor in their approach, but never define nor describe what these are.
- The paper proposes cDFAs as a novel goal-representation for temporal tasks in goal-conditioned RL. Beyond the fact that I am not convinced that they are meaningfully different from DFAs, the paper gives no theory nor conducts any experiment to demonstrate that they "balance formal semantics, accessibility, and expressivity" as claimed in their first contribution. In general, since the authors claim it as a main contribution of this work, they should justify why it is better than other representations like RMs, LTL, GTL, TLTL, SPECTRL, etc. I think such rigorous justifications are not needed if they did not claim it as a contribution of this work.

### Experiments
- It is not clear why the main paper contains no experiment comparing with prior works. The only such experiment is left to the appendix, and only uses LTL2Action as baseline which is not state-of-the-art (there is also no sample-efficiency comparison). Without state-of-the-art baselines like [1] or [2] in the main paper, it is hard to tell how good the proposed approach is. Given that [2] drastically outperforms LTL2Action in their experiments in the same domains, this work does not look promising as it only shows similar performance to LTL2Action.
- The paper claims that the proposed approach is applicable to any DFA, but the DFAs used for pretraining the goal encoder (Algorithm 1) and the ones used for the experiments (page 7) only have one atomic proposition per edge (e.g. $red$, $square$, etc). These are the simplest DFA transitions possible. In general the edge of a DFA is a Boolean expression over the atomic propositions (e.g. $red \wedge \neg square$). It is unclear if the approach generalises to more complex DFAs (where the edges/transitions can be arbitrary Boolean expressions).
- All the example DFAs used are generally not satisfiable in the environments considered since their states lack self-transitions (e.g. in Fig 1). Since all the task distributions used for the experiments are defined similarly, it is unclear why the results show high success rates on them (they should be generally unsolvable). This suggests that important details of the implementation of the approach for the experiments may be different from what has been described in the paper.
- It is also unclear what the tasks sampled from the defined distributions look like. It would have helped if the authors included examples of the simplest and most complex DFAs sampled during training and during testing.
- It is unclear why freezing the pretrained weights performs better than finetuning. This is counter-intuitive but is never explained in the paper.
- Given how the performance of the proposed approach is heavily reliant on the goal embeddings obtained from pretraining (the performance is abysmal without it), it is concerning that the paper did not stress test it to understand when it fails and why. For example, what if the number of conjunctions goes up to 100 where each DFA has a task length that also goes up to 100 (relates to the temporal curse of dimensionality)? What if the DFA transitions are more complex, that is when they are arbitrary Boolean expressions over the atomic propositions (relates to the spatial curse of dimensionality)?

## Significance
- This work does not make a very large step forward from prior work (the addition of state-aware planning), but it is one. So I do not want to be overly pessimistic on the front of significance, but it is necessary to note that the contributions of this work are relatively incremental. If I understood the paper correctly, the only main contribution is the approach for pretraining a *good* goal (DFA) embedding. Additionally Figures 20-21 also only shows similar performance to non-state-of-the-art prior work (LTL2Action).
- The absence of theory does not help. Theory guaranteeing generalisation and convergence to optimal policies (as claimed) under some reasonable assumptions would have helped. The lack of experiments comparing with state-of-the-art baselines like [1,2] also does not help.
- Relying only on a UVFA for generalisation has clear limitations (e.g. in terms of the spatial and temporal curses of dimensionality present in temporal logic tasks), since no optimality and generalisation guarantees can be given even in finite MDPs (like grid-worlds). This is why the field has largely moved away from such approaches. More recent works such as [1,2] (and the large majority of prior works like [3,4]) break down the problem into a set of reach-avoid sub-tasks (goals), then use planning (or HRL) to determine how to sequence those goals (addressing the temporal curse). These works use UVFA for the sub-goal policies only when the goal space is sufficiently large. When the goal space isn't too large, such as in the letter grid-world, these methods can still learn and generalise without relying on a UVFA (by learning the policies/options for each goal independently).
- This reliance on end-to-end function approximation for generalisation hurts the significance of this work.


[1] Tasse, Geraud Nangue, et al. "Skill Machines: Temporal Logic Skill Composition in Reinforcement Learning." The Twelfth International Conference on Learning Representations. (2024)

[2] Qiu, Wenjie, Wensen Mao, and He Zhu. "Instructing goal-conditioned reinforcement learning agents with temporal logic objectives." Advances in Neural Information Processing Systems 36 (2024).

[3] Araki, Brandon, et al. "The logical options framework." International Conference on Machine Learning. PMLR, 2021.

[4] Jason Xinyu Liu, Ankit Shah, Eric Rosen, George Konidaris, and Stefanie Tellex. Skill transfer for temporally-extended task specifications. arXiv preprint arXiv:2206.05096, 2022.

# Minor

- It is not clear what "concept class" means or how it is different from a "task class". Are they the same?
- It is not clear what GATv2 in the caption of Figure 1 means. Please add in the caption that it is a GNN.
- It is seems like "task" and "goal" are mean the same thing in this work (line 16), i.e. they mean a DFA. Is that correct? That wasn't clear.
- Adding the cDFA in Fig 1 to Fig 2 will
- There are a couple minor typos or writing mistakes. For example:
 - There are numerous places where shorthands like "i.e." and "e.g." are used (e.g. first paragraph of the introduction). It is not recommended in a formal writing. Please spell them out.
 - The sentence on Line 184 doesn't end.

**Questions:**

Please refer to the weaknesses above.

**Limitations:**

The authors have discussed some of the limitations. However, there seems to be several other ones. For example,

- Just like *some* prior works, the paper assumes that all tasks are defined over a fixed set of given atomic propositions. If the set increases, everything needs to be retrained.
- The paper demonstrates their approach only on DFAs with one atomic proposition per edge. It is unclear how it will behave when the DFA transitions are more complex (i.e. arbitrary Boolean expressions over the atomic propositions)?
- The paper relies on UVFAs to learn general policies. Hence, it is likely that it can only generalise to tasks *similar* to those it is trained on, and not to tasks that are *significantly* out of distribution. The paper does not investigate when the model fails to generalise and why. That would have helped the reader have some justified intuition of what types of tasks are *significantly* out of distribution for function approximation to fail for proposed approach.


I recommend the authors carefully think of the various significant limitations of their approach, such as the one I mentioned above, and explicitly state and discuss them in the paper.

---

> ### Author Rebuttal · Authors · 2024-08-06
>
> _Thank you for the time and effort put into your review._
>
> ### Regarding LTL2Action not being finite
> The reviewer is right in the sense that in the LTL2Action paper, the encoder could *mechanically* be applied to standard LTL, but there is no indication that this will work. **The reality is that the training code and experiments only consider finite LTL.**
> For example the [corresponding line in the LTL2Action repo](https://github.com/LTL2Action/LTL2Actionblob/485cbc1055dc9fbfbba50350c85de11fc1730540/src/ltl_wrappers.py#L98) shows that an LTL task is declared ``done'' when the underlying LTL formula cannot be ``progressed'' anymore. The LTL2Action paper leaves room for the idea
> of using rewards to encode infinite horizon tasks, but we note that this is an active research topic with some of the more [theortically sound and practical](https://proceedings.mlr.press/v202/voloshin23a.html) works on the subject post-dating LTL2Action. As such, we stand by our initial assement that LTL2Action should be compared against techniques for finite trace properties.
>
> ### DFAs vs cDFAs
> Please see the global rebuttal for an explanation of extension to negations and disjunctions.
>
> ### Comparison with GCRL-LTL
> Although the solved problems are similar, comparison with GCRL-LTL is an ``apples to oranges'' comparison as it is a hierarchical planning-based method that does not generate embeddings. These are fundamentally different approaches
> and the work on GCRL-LTL already provides a comparison. Such approaches will, of course, learn faster than an embedding space-based approach since they only learn the goals in the environment, and the temporal planning is done algorithmically. However, as pointed out by Reviewer bN73, we believe there are many interesting future works that could follow from our approach. Please see section **Baselines** in the global rebuttal for more details.
>
> ### Having Boolean expressions on edges
> Standard DFAs cannot have Boolean expressions on edges. Automata with Boolean expressions on edges are variants of symbolic automata, which is an extension that we leave for future work.
>
> ### Regarding self transitions
> As written on line 137 of the submission, for ease of notation, we assume _stuttering semantics_: "Omitted transitions are assumed to stutter, that is transition back to the same state".
>
> ### Why is freezing better?
> This is a good question. Our current hypothesis is that the RAD pre-training learns a sufficently good representation that introducing changes to it and/or computing the corresponding gradients ultimately makes performance worse. In particular, less of the gradient updates are directed at learning dynamics and instead try to overfit to the specific concept class.
>
> However, as this is pure speculation, we have left such discussion out of the paper and think its an interesting avenue for future research.
>
> ### Stress testing
> The issue with the proposed conjunction stress test is that the more DFA you have in a conjunction, the less likely that the sampled task is satisfiable since each DFA is essentially a constraint over the set of accepted behaviors, and as the number of constraints increases, it is more likely to end up with an empty set. So it is hard to say if the framework would break because of a limitation of the approach or because the sampled tasks are not satisfiable. Nevertheless, its an interesting question to design a stress test that approriately seperates the the failure causes.
>
> ### Comments on end-to-end function approximation
> We respectfully disagree that the community is moving away from end-to-end function approximation, particularly for reward specification. If anything, the introduction of strong multi-modal image/language embeddings such as CLIP have resulted in an explosion of work focused on end-to-end learning from embeddings of a task. They include using [sketches](https://rt-sketch.github.io/), [language corrections](https://yay-robot.github.io/), and of course [language instruction](https://llarva24.github.io/).
>
> Finally, while hierarchical planning is a powerful approach, as previously discussed, it is not a complete replacement for end-to-end function approximation. We believe that a focus on function approximation and embeddings for logical specifications not only aids generalization but also provides a path forward to integrate multiple modes (DFA embeddings, natural language embeddings, image embeddings) in machine learning models.
>
> ### On exposition and definitions
> Thank you for your feedback. As reviewer WW4E noted, combining formal objects like DFAs and MDPs can result in notation-heavy and unapproachable work. One technique we employed in the definitions of MDPs and goal-conditioned RL was a simplification of the essential components necessary to understand the material at hand. We will use the additional space afforded by the camera ready to remark that these are simplifications of the more general definitions as you note.
>
> For example, we indeed omitted the reward (as is sometimes done in inverse learning works) to emphasize that we are in the binary sparse reward setting. Similarly, because goal-conditioned RL reduces to maximizing reach probability, this is the definition we employ. Again, we will make this clear in future versions.
>
> Finally, we chose to omit definitions such as message passing NN and employ short hands such as "e.g.", "i.e." as they are standard within the NeurIPs community.
>
> - **Regarding Defn 3.1.** DFA-condition policies use both the environment state and the DFA embeddings. This pair corresponds to the augmented state space. We showed this by abusing the notation and writing $\pi : S \times DFA$. We will make it clear in the final version.
> - **Minor questions.** We use "concept class" and "task class" as well as "goal" and "task" interchangeably. We will clarify this and your other comments in the final version.

---

> > ### Comment · Reviewer_A6oy · 2024-08-14
> > **Comment by Reviewer A6oy**
> >
> > Thank you to the authors for their detailed response and clarifications. I particularly appreciate the comments regarding the originality and significance of the proposed approach, and the clarifications regarding DFAs vs cDFAs. The intuition behind why frozen embeddings are giving the best results is also interesting. I think a detailed discussion of this will improve the quality of the paper, given that it is the second main contribution of the paper.
> >
> > I believe the RAD pretraining approach is interesting and as I mentioned in my original review, it could be a useful contribution to the field. But I am not fully convinced by the provided emperical evidence. I have the following main outstanding concerns regarding this paper:
> >
> > - GCRL+LTL/DFA and end-to-end function approximation (UVFA-only) methods do have their tradeoffs. As the authors have highlighted, GCRL based methods are often very sample-efficient but often suffer from suboptimality which requires planning while taking into account the environment states (e.g. using Dijkstra). Similarly, UVFA-only methods have the potential to learn optimal policies, but lack any optimality and generalisation guarantees due to the reliance on function approximation. Hence, it is important to understand emperically or theoretically when a proposed UVFA-only approach is likely to suffer from sub-optimality. My main concern here is that there's not enough investigation done on the tradeoffs induced by the RAD pretraining approach, and the reliance on UVFAs for generalisation over cDFAs. Such investigations are particularly important in RL research where the statistical significance of results is often very low (for example this paper shows results over only 10 seeds, which is common, but likely statistical insignificant).
> >
> > - Regarding DFAs with Boolean expressions on edges, I meant Boolean expressions over atomic propositions (AP). I.e. when your set of symbols is $2^{AP}$, as is commonly done. The experiments only have edges over AP, which is a severely restricted class of DFAs.
> >
> > - I see what you mean for the conjunction stress test. But you can still define such tasks. An example in the letter world is the conjunction of all 12*11 unique SRA tasks: Reach Letter$\_i$ while avoiding Letters$\_{j \neq i}$ . You can also have a sequencial stress test. E.g. SRA task of length k ∼ Uniform(1,100).
> >
> > -  As I mentioned previously, it will help to include examples of the simplest and most complex DFAs from the defined distributions.
> >
> > I have increased my score to reflect the clarifications provided and my outstanding concerns.

---

### Author Rebuttal · Authors · 2024-08-06

_We'd like to thank the reviewers for their time and efforts.
Below, are common points we'd like to emphasize across all reviews._

## RAD Pretraining + Frozen Embeddings
First, we wanted to highlight what we consider the most important contributions of our work:

1. The RAD pretraining.
1. The frozen encoder and RAD embeddings.

Notably, RAD is a carefuly crafted task distribution
that exploits DFA specific structure to encourage zero-shot generalization to downstream tasks.
Reviewer bN73 put it well when they wrote:

> [RAD is designed from] what motifs ought to be repeated across a large class of DFAs (namely, reach-avoid tasks), and how that could be exploited to enable generalization.

We believe that our generalization results (Figure 6 and Appendix C.2-C.8), training with frozen embeddings (Figure 4), and embedding visualizations (Figure 5 and C.3) speak to the usefulness and generality of RAD pretraining. *We see RAD pretraining as a building block for other applications.*

- **Multi-modal task descriptions:** One might consider combining multiple embeddings for control problems. For example using our RAD embeddings to specify a mission level task, preferences using natural language CLIP embeddings, and a grounding of atomic propositions/labels using [hand drawn sketchs](https://rt-sketch.github.io).
- **Task retrevial:** In the future, one might train a natural language to RAD embeddings network to enable vector database retrieval of tasks that a robot is pre-authorized to perform.


## Baselines
A common question raised by reviews (A6oy, 7HER) is about baselines. The only existing approach directly comparable to ours is LTL2Action, which we compare against in Appendix C.8. With the additional space, we will move this comparison to the main body of the paper.

Several other works (e.g., [1,2,3,4] from reviewer A6oy and GCRL-LTL also mentioned by 7HER) pursue a totally different technique for goal-conditioning with logical specifications. These works use a hierarchical framework: classical planning techniques (e.g. Dijkstra’s algorithm) are used to plan over an automaton (or automaton-like structure) and trained neural policies accomplish the individual steps in the plan. While these works can also be used for satisfying logical specifications over MDPs, they differ in two critical ways:

1. They don't produce task embeddings. Our work is explicitly focused on generating useful task embeddings (see above for why), which we accomplish largely through RAD pretraining.
2. Hierarchical planning techniques (including all of the papers mentioned above) are generally sub-optimal. Figure 1 in the attached pdf shows a gridworld demonstrating the problem with hierarchy. The robot's task is to first go to orange and then green. A hierarchical plan would first attempt to reach orange in the quickest way, which would be going to square (a). However, because hierarchical plans are unable to account for the entire task and the dynamics of the MDP, they don't realize that this now takes them further away from the second part of the task, reaching green.The situation can be made substantially worse if (i) the non-Markovian reasoning requires reasoning even further into the future and (ii) the dynamics of the MDP were somehow irreversible so that choosing the wrong orange square makes it _impossible_ to then reach the green square. It is easy to construct such traps for current hierarchical approaches. By using task embeddings (our work and LTL2Action), policies reason about the _full_ task, and thus could avoid such traps.

We will make these differences clearer in the final draft.

## Differences from LTL2Action

As noted in the paper and by several other reviewers, the mechanics of our approach take inspiration from LTL2Action.
Superficially, the main changes are the encoder (now based on GATv2) and the target representation (automata rather than logical sentences).

A key argument of the paper is that by focusing on DFA, we can exploit the fact that (i) solving a DFA ``just'' corresponds to finding a satisfying path and (ii) that subproblem can itself be represented as a DFA. This is fundamentally not the case with **declarative semantics** such as temporal logic which are naturally represented as syntax trees and require non-local reasoning (does satisying one part of the subtree effect the satisfaction of the other?) As the next section emphasizes, cDFA offer a middle ground and support the non-temporal syntatic structure of the LTL while allowing DFAs to handle the temporal relations.


## cDFAs.
The conjunctive composition (cDFAs) investigated in the paper joins the graphs of several DFAs into a single graph. This is done by adding an "AND" node that is connected to each DFA in the composition and avoids explicitly computing the composition of the DFAs (an exponential blow-up of the number of states). For example, given N DFAs each with M states, their conjunctive composition is a DFA with $O(M^N)$ many states whereas their cDFA representation (which does not compute the composition explicitly) has $O(M\cdot N)$ many states. The cDFA representation avoids *the exponential blow-up in the number of states caused by the explicit evaluation of the composition*; therefore, addresses the curse of dimensionality.

**Arbitrary DFA compositions.** One can express arbitrary Boolean combinations of DFAs in a cDFA representation, using a CNF tree. Extending conjunctive composition to arbitrary Boolean combinations follows from three observations:

1. Any Boolean formula can be expressed in the CNF format.
2. We can combine DFAs in a CNF tree, where the "AND" node would again be the root node, i.e., level 0, the disjunction nodes would be in level 1, and DFAs would be in the leaves.
3. DFAs are closed under negation, and the number of states stays constant when a DFA is negated, so negations can be directly applied to individual DFAs.

---

### Decision · Program_Chairs · 2024-09-25

**Decision:**

Accept (poster)

**Comment:**

The paper proposes a method for goal-conditioned reinforcement learning using a collection of deterministic finite automata (cDFAs) to define temporally-extended tasks. It introduces a graph neural network to embed cDFAs into vectors and a pretraining procedure using a Reach Avoid Derived (RAD) task distribution. This pretraining improves the generalization and performance of policies by leveraging these embeddings. Experiments demonstrate that freezing the pretrained embeddings results in better performance and showcases generalisation to several task classes.

There was much discussion on the paper, in particular with the positioning of the proposed approach in relation to existing work (most notably LTL2Action). This was addressed to the satisfaction of the reviewers during the rebuttal phase. There are still some outstanding issues with the paper: more empirical evaluations comparing against GCRL-LTL and understanding the tradeoffs between this style of approach and one involving UVFAs only, would strengthen the paper, and there are still concerns about how the RAD pretrained embeddings degrade depending on the task distribution.

Despite these shortcomings, the reviewers generally agree on acceptance, and authors are encourage to incorporate additional experiments and suggestions as provided by the reviewers.